# Caspr1 is a host receptor for meningitis-causing *Escherichia coli*

Wei-Dong Zhao[1], Dong-Xin Liu[1], Jia-Yi Wei[1], Zi-Wei Miao[1], Ke Zhang[1], Zheng-Kang Su[1], Xue-Wei Zhang[1], Qiang Li[1], Wen-Gang Fang[1], Xiao-Xue Qin[1], De-Shu Shang[1], Bo Li[1], Qing-Chang Li[2], Liu Cao[1], Kwang Sik Kim[3] & Yu-Hua Chen[1]

*Escherichia coli* is the leading cause of neonatal Gram-negative bacterial meningitis, but the pathogenesis of *E. coli* meningitis remains elusive. *E. coli* penetration of the blood–brain barrier (BBB) is the critical step for development of meningitis. Here, we identify Caspr1, a single-pass transmembrane protein, as a host receptor for *E. coli* virulence factor IbeA to facilitate BBB penetration. Genetic ablation of endothelial Caspr1 and blocking IbeA–Caspr1 interaction effectively prevent *E. coli* penetration into the brain during meningitis in rodents. IbeA interacts with extracellular domain of Caspr1 to activate focal adhesion kinase signaling causing *E. coli* internalization into the brain endothelial cells of BBB. *E. coli* can invade hippocampal neurons causing apoptosis dependent on IbeA–Caspr1 interaction. Our results indicate that *E. coli* exploits Caspr1 as a host receptor for penetration of BBB resulting in meningitis, and that Caspr1 might be a useful target for prevention or therapy of *E. coli* meningitis.

[1] Department of Developmental Cell Biology, Key Laboratory of Cell Biology, Ministry of Public Health, and Key Laboratory of Medical Cell Biology, Ministry of Education, China Medical University, 77 Puhe Road, Shenbei New District, 110122 Shenyang, China. [2] Department of Pathology, China Medical University, 77 Puhe Road, Shenbei New District, 110122 Shenyang, China. [3] Division of Pediatric Infectious Diseases, Johns Hopkins University School of Medicine, 200 North Wolfe St, Room 3157, Baltimore, MD 21287, USA. These authors contributed equally: Wei-Dong Zhao, Dong-Xin Liu. Correspondence and requests for materials should be addressed to W.-D.Z. (email: wdzhao@cmu.edu.cn) or to Y.-H.C. (email: yhchen@cmu.edu.cn)

Neonatal bacterial meningitis is associated with substantial mortality and morbidity. Despite advances in intensive care, survivors sustain neurologic sequelae such as hearing loss, developmental delay and cognitive impairment[1, 2]. *Escherichia coli* (*E. coli*) is the most common cause of Gram-negative meningitis in newborn infants[3], and most cases of *E. coli* meningitis occur as the consequence of hematogenous spread[4, 5].

Invasion of the circulating *E. coli* into the central nervous system (CNS) is the critical step for development of meningitis, which requires bacterial penetration through the blood–brain barrier (BBB)[6]. BBB, a selective semi-permeable barrier that separates the brain from circulating blood, is primarily composed of brain microvascular endothelial cells (BMECs), basal lamina, astrocyte end-feet and pericytes[7]. During the past decades, in vitro cultured BMECs have been developed to study the invading mechanisms of CNS-infecting pathogens through the BBB. In *E. coli*, several virulence factors associated with bacterial invasion of BMECs have been characterized, including Ibe proteins[8–11], OmpA[12], CNF1[13], and FimH[14]. The subsequent studies have attempted to identify the potential host receptors in BMEC for these bacterial factors. In vitro binding of *E. coli* virulence factors with the surface molecules in BMECs were reported; e.g., CNF1 interacts with 37-kDa laminin receptor precursor (37LRP)/67-kDa laminin receptor (67LR)[15, 16]. However, the physiological roles of these interactions in the context of bacterial meningitis remain to be established. Identification of the host receptors recognizing *E. coli* virulence factors is still an open question.

Following breaching the BBB, the pathogenic bacteria will gain access to CNS, resulting in intracranial inflammation and neuronal injury. Neuronal cell death has been described in bacterial meningitis[17–19], which could account for the neuropsychological deficits in the survivors. In the animal models of *E. coli* meningitis, apoptosis of neurons was present in the dentate gyrus of the hippocampus[20]. Evidence from experimental meningitis caused by other CNS-infecting pathogens, such as *Streptococcus pneumoniae*[17, 21–23] and group B *Streptococcus*[18, 24] indicated that apoptosis is a major contributor to neuronal cell death. The molecular mechanism of neuronal apoptosis in bacterial meningitis remains poorly understood.

Caspr1 (contactin-associated protein 1, also known as paranodin), a single-pass transmembrane protein, was originally identified in neurons by its interaction with contactin protein[25, 26]. Caspr1 is localized in the paranodal region and has a role in the generation and maintenance of paranodal junctions in myelinated axons[25–28].

Here, we reveal that Caspr1 is expressed in the brain endothelium of rodents, and Caspr1 acts as a receptor of bacterial virulence factor IbeA for *E. coli* penetration through the BBB. In addition, we show that neuronal Caspr1 is exploited by IbeA for the *E. coli* invasion into neurons leading to neuronal apoptosis. Inhibition of endothelial Caspr1 function significantly attenuates pathogenesis of *E. coli* meningitis by preventing *E. coli* crossing the BBB. Our study presents Caspr1 as a potential target for intervention of *E. coli* meningitis.

## Results

**Brain endothelial Caspr1 interacts with bacterial IbeA**. IbeA is a critical determinant in the meningitis-causing *E. coli* contributing to bacterial penetration through the BBB[9, 10]. Given that IbeA protein could be secreted from *E. coli* into the extracellular media upon contact with brain endothelial cells (Supplementary Fig. 1A, B), we performed yeast two-hybrid to identify the interacting partners of IbeA in human brain microvascular endothelial cells (HBMECs), which is the primary component of the BBB. IbeA was used as bait to screen HBMECs cDNA library,

and four clones encoding Caspr1 were obtained (Supplementary Table 1). Yeast cells co-transformed with the bait vector (pGBK) containing IbeA and prey vector (pGAD) containing Caspr1 were able to grow and form blue colonies on the selection plates, suggesting IbeA interacted with Caspr1 (Fig. 1a). The full-length cDNA of Caspr1 was successfully cloned from HBMECs. The cell lysates of human embryonic kidney (HEK) 293T cells transfected with full-length His-tagged Caspr1 were incubated with Glutathione Sepharose 4B beads prebound with GST-IbeA or GST (as control), and the following western blot revealed the binding of IbeA with exogenous expressed Caspr1 (Fig. 1b). Furthermore, cell lysates of HBMECs were incubated with the beads prebound with GST-IbeA, and the western blot results showed that endogenous Caspr1 in HBMECs interacted with IbeA protein, but not with non-IbeA proteins such as OmpA (outer membrane protein A)[29], another virulence factor involved in *E. coli* meningitis (Fig. 1c). Subcellular fractionation analysis revealed the localization of Caspr1, a single-pass transmembrane protein, in the membrane fraction of HBMECs (Fig. 1d). Caspr1 was localized at the plasma membrane and perinuclear region in HBMECs, and co-localized with VE-cadherin, a plasma membrane marker (Fig. 1e). Immunostaining of rat brain slices revealed the co-localization of Caspr1 with CD31 (a marker for endothelial cells), indicating Caspr1 was present in brain microvessels (Fig. 1f). In contrast, the vascular expression of Caspr1 in other tissues (e.g. liver, kidney and lung) was not detectable (Supplementary Fig. 1C). Furthermore, immunoelectron microscopy confirmed Caspr1 localization at the luminal side of brain microvessels (Fig. 1g, Supplementary Fig. 1D). These results indicated that Caspr1, an integral membrane protein present in brain microvessels, interacts with *E. coli* virulence factor IbeA.

**Caspr1 is required for *E. coli* penetration through the BBB in vivo**. The mice carrying *loxP*-flanked *Caspr1* allele (*Caspr1^{loxP/loxP}*) were generated in which two *loxP* sites were inserted before exon 4 and after exon 1 of *Caspr1* gene, respectively (Fig. 2a). *Caspr1^{loxP/loxP}* mice were crossed with VE-cadherin-Cre mice (*VE-cadherin^{Cre/+}*) in which expression of Cre-recombinase is under the control of the endothelial-specific VE-cadherin promoter[30], yielding *Caspr1^{loxP/+}; VE-cadherin^{Cre/+}* mice. Then the *Caspr1^{loxP/+}; VE-cadherin^{Cre/+}* mice were bred with *Caspr1^{loxP/loxP}* to produce *Caspr1^{loxP/loxP}; VE-cadherin^{Cre/+}* mice in which *Caspr1* was knocked out in endothelium (Caspr1 eKO). The resulting transgenic mice carrying the *loxP* and *Cre* were identified by PCR genotyping (Supplementary Fig. 2A). Immunostaining results showed that expression of Caspr1 was absent in brain microvessels in Caspr1 eKO mice (Fig. 2b) indicating successful deletion of Caspr1. Then we tested whether the occurrence of bacterial meningitis is affected in neonatal Caspr1 eKO mice with experimental hematogenous *E. coli* meningitis. The mice were injected with wild-type *E. coli* strain subcutaneously to induce meningitis. Then the blood specimens were collected and cultured for demonstration of bacteremia. The results showed that the levels of bacteremia were similar in littermate control (*Caspr1^{loxP/loxP}*) mice and Caspr1 eKO mice (Fig. 2c). In the meantime, the cerebrospinal fluid (CSF) were collected and cultured to indicate the passage of bacteria through the BBB, and the inflammation of the cerebral meninges were determined by histological examination of the brain sections indicated as meningeal thickening and neutrophils infiltration in the meninges. Interestingly, we found the rate of meningitis occurrence, defined as positive CSF culture with meningeal inflammation, was reduced to 25% in Caspr1 eKO mice ($n = 16$), which is significantly lower than the 61% in control mice ($n = 18$) ($P < 0.05$, $\chi^2$ test, Fig. 2c, d). Consistently, when neonatal rats

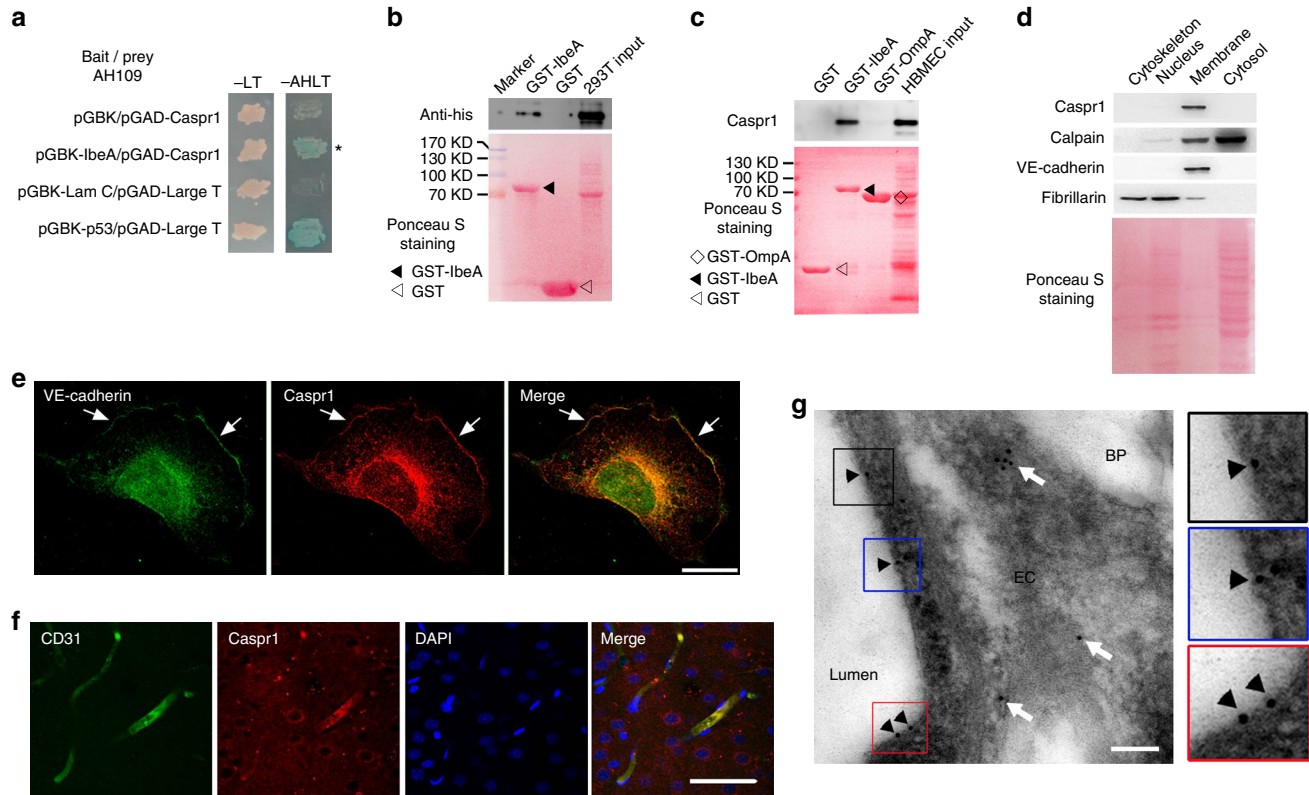

**Fig. 1** Endothelial Caspr1 is identified as the binding protein of bacterial IbeA. **a** Yeast cells co-transformed with bait and prey plasmids were grown on SD/-Leu-Trp (-LT) selection medium and SD/-Ade-His-Leu-Trp (-AHLT) medium containing X-α-Gal. Co-transformation of Lamin C and Large T served as a negative control, whereas p53 and Large T was a positive control. **b** HEK293T cells were transfected with His-tagged full-length Caspr1, then the cell lysates were prepared and co-incubated with the beads prebound with GST-IbeA for GST pulldown assay, with GST as control. The precipitates were analyzed by western blot using His antibody (upper). Full blots are shown in Supplementary Fig. 8. The blotting membrane was stained with Ponceau S as a loading control (lower). **c** HBMECs were lysed and co-incubated with the beads prebound with GST-IbeA for GST pulldown assay, with GST and GST-OmpA as controls. The precipitates were analyzed with Caspr1 antibody (upper). Full blots are shown in Supplementary Fig. 8. The blotting membrane was stained with Ponceau S as a loading control (lower). **d** Subcellular components of HBMECs were extracted and analyzed by western blot with antibodies recognizing Caspr1, Calpain (cytosolic marker), VE-cadherin (membrane marker), and Fibrillarin (nucleus marker), respectively (upper). Full blots are shown in Supplementary Fig. 8. The blotting membrane was stained with Ponceau S as a loading control (lower). **e** HBMECs were fixed and immunofluorescence was performed with antibodies recognizing Caspr1 (Red) and VE-cadherin (Green). Arrows indicate the co-localization of Caspr1 with VE-cadherin. The expression of VE-cadherin in the nuclear fraction was undetectable in (**d**), thus we consider the fluorescence signals of VE-cadherin in the nucleus might be caused by non-specific staining. Scale, 50 μm. **f** The rat brain slices were prepared and stained with Caspr1 (Red) and CD31 (Green) antibodies. DAPI was used to identify nuclei (blue). Scale, 50 μm. **g** Ultra-thin sections were prepared from the rat brain and labeled with primary antibody recognizing Caspr1, followed by incubation with secondary antibody conjugated with 10-nm gold particles. Images were acquired using transmission electron microscope. EC: endothelial cell, BP: brain parenchyma, Lumen: vascular lumen. Arrowheads indicate labeling of Caspr1 in the luminal side of endothelial cell whereas arrows indicate intracellular labeling. Scale, 100 nm. The zoom-in windows of the positive labeling of the luminal Caspr1 were provided (right). All the images are representative of three independent experiments

were intraperitoneally injected with the antibody recognizing extracellular region of Caspr1 to block the Caspr1 on brain microvessels, the rate of positive CSF was decreased ($P < 0.05$, Supplementary Fig. 2B, C). Further results revealed that the survival of mice inoculated with *E. coli* were prolonged in Caspr1 eKO mice compared to control mice ($P < 0.05$, Fig. 2e). These data indicated that brain endothelial Caspr1 is associated with *E. coli* penetration through the BBB.

*E. coli* penetration through the BBB occurs via a transcellular mechanism[6]. The internalization of *E. coli* into host brain endothelial cells is dependent on two major processes, adhesion and invasion. To further determine the mechanism of Caspr1-mediated infection, we used RNA interference to knockdown Caspr1 in HBMECs (bottom panel, Fig. 2f) followed by *E. coli* adhesion and invasion assays. Knockdown of Caspr1 did not affect the viability of HBMECs (Supplementary Fig. 2D). We found Caspr1 knockdown in HBMECs significantly reduced

internalization of *E. coli* into HBMECs to ~41% of the control (bottom panel, Fig. 2f), whereas *E. coli* adhesion to HBMECs was not affected (top panel, Fig. 2f). The reduced bacterial invasion was effectively rescued by full-length Caspr1 cDNA (Fig. 2g). Moreover, when HBMECs were pre-incubated with antibody recognizing Caspr1's extracellular region to block the Caspr1 on membrane surface, *E. coli* invasion was decreased in a dose-dependent manner compared to IgG control (bottom panel, Fig. 2h), whereas Caspr1 antibody did not affect *E. coli* adhesion to HBMECs (top panel, Fig. 2h). In contrast to these results, knockdown of Caspr3, another member of the Caspr family that is highly expressed in HBMECs (Supplementary Fig. 2E), did not affect *E. coli* invasion into HBMECs (Supplementary Fig. 2F). These data demonstrated that host Caspr1 is specifically associated with *E. coli* invasion into brain endothelial cells to facilitate bacterial penetration through the BBB during meningitis.

**Extracellular domain of Caspr1 interacts with bacterial IbeA**.
The region of Caspr1 bound with IbeA in yeast two-hybrid assays
corresponded to the N-terminal laminin-globular (lam-G)
domain (Fig. 3a, Supplementary Fig. 3A), one of the four major
structural domains in Caspr1. It is therefore proposed that one of
the lam-G domain in Caspr1 might be responsible for the
interaction between Caspr1 and IbeA. To test this, truncated
forms of Caspr1 were constructed based on the predicted domain
architecture (Fig. 3a), without disrupting the lam-G domains. The
resulting constructs were translated using in vitro transcription/

translation system and subjected to GST pulldown assays to
determine which domains in Caspr1 could bind GST-tagged IbeA
protein. The results showed that aa 203–538 of Caspr1, con-
taining two N-terminal lam-G domains, bound with IbeA
(Fig. 3b). Then the truncated constructs were transiently trans-
fected into HEK293T cells for protein expression, and the cells
were lysed and subjected to GST pulldown assay. The results
showed that aa 203–538 of Caspr1 indeed interacted with IbeA
protein (Fig. 3c). In order to determine the minimum domain in
Caspr1 critical for the interaction, two constructs containing aa

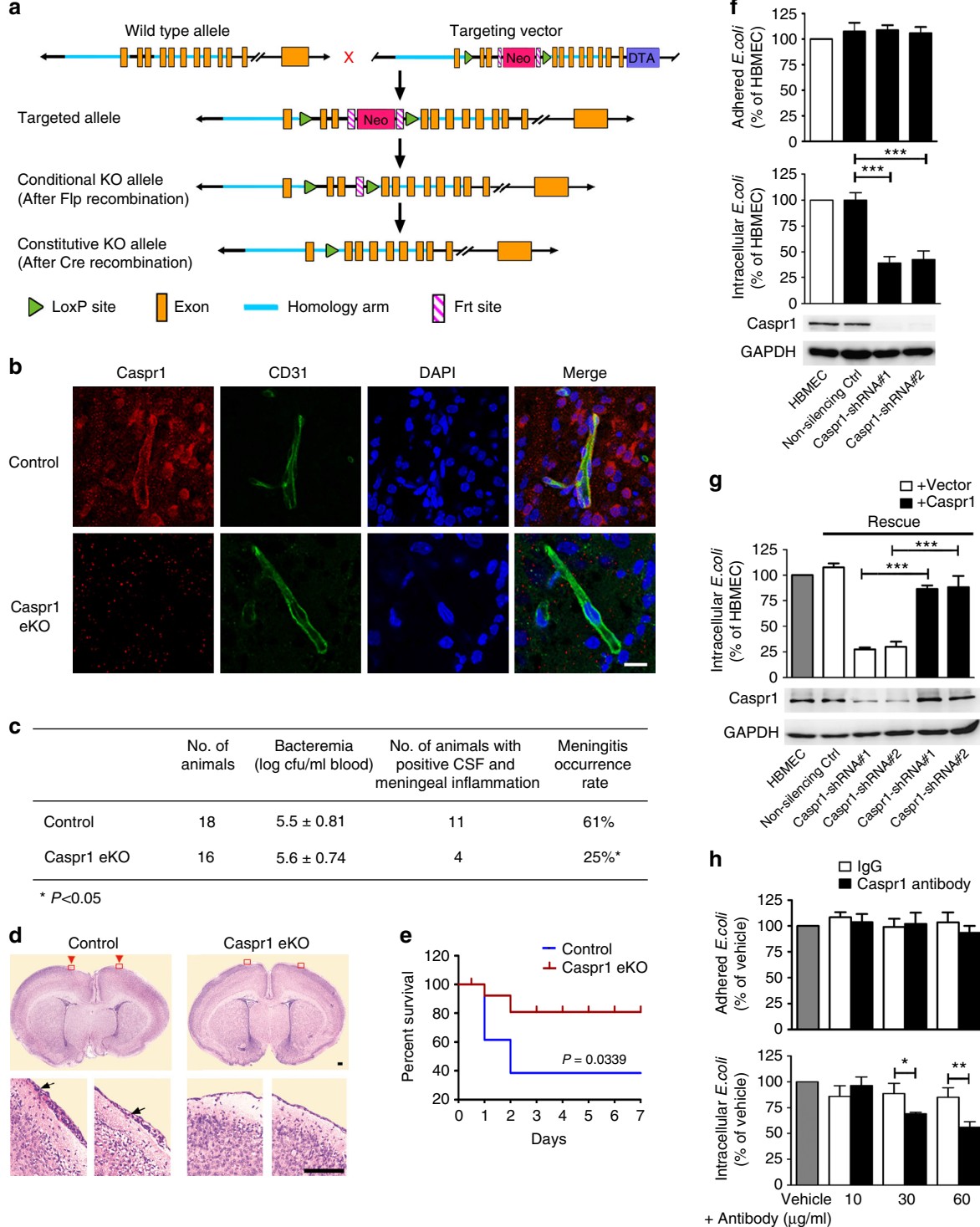

203–355 and aa 356–538 of Caspr1, each of which encoding a single lam-G domain, were prepared for in vitro transcription/translation followed by GST pulldown assay. The results indicated that aa 203–355 of Caspr1, containing the first N-terminal lam-G domain, interacted with IbeA (Fig. 3d). Then the HBMECs stably expressing Caspr1 mutant lacking residues 203 to 355 (Caspr1Δ203–355) were established (bottom panel, Fig. 3e) and the localization of Caspr1Δ203–355 at the plasma membrane were verified (Supplementary Fig. 3B). The Caspr1Δ203–355-expressed HBMECs were assessed by bacterial invasion and adhesion assays and the results showed that expression of Caspr1Δ203–355 mutant reduced E. coli invasion into HBMECs (bottom panel, Fig. 3e), whereas the adhesion of E. coli to HBMECs remained unaffected (top panel, Fig. 3e). These results indicated that aa 203–355 of Caspr1, the N-terminal lam-G domain showing limited similarity (40%) with the other 3 lam-G domains in Caspr1 (Supplementary Fig. 3C), specifically interacted with E. coli IbeA to facilitate bacterial internalization into the brain endothelial cells.

Next, the domains in IbeA interacting with aa 203–355 of Caspr1 were determined by preparing truncated forms of IbeA tagged with GST (Fig. 3f) and the binding of IbeA with in vitro translated His-tagged Caspr1(203–355) was examined. The results showed that Caspr1(203–355) bound with the aa 229–342 of IbeA (Fig. 3g). Then an E. coli mutant strain with deletion of aa 229–342 in IbeA, E44-IbeAΔ(229–342), was constructed by isogenic in-frame deletion in parent wild-type E. coli strain (E44) (Fig. 3h) and the secretion of IbeAΔ(229–342) mutant protein were verified (Supplementary Fig. 3D). The results from bacterial invasion and adhesion assays showed that, compared to wild-type E44, the ability of E44-IbeAΔ(229–342) to invade HBMECs was significantly decreased (lower panel, Fig. 3i), comparable to that of full-length ibeA deletion mutant strain, ZD1[9]. The bacterial adhesion remained unaffected in E44-IbeAΔ(229–342) strain (top panel, Fig. 3i). The decreased invasion in ZD1 strain was effectively rescued by full-length ibeA gene, but not by ibeA lacking aa 229–342 (Fig. 3j). These results demonstrated that the 229–342 domain of E. coli IbeA interacted with the 203–355 domain of endothelial Caspr1 to promote E. coli invasion into brain endothelial cells.

**IbeA–Caspr1 interaction activates host cell FAK-Rac1 signaling.** Then we investigated the intracellular signaling in brain endothelial cells exploited by IbeA–Caspr1 interaction. Pulldown experiments followed by mass spectrometry suggested that focal adhesion kinase (FAK), the activation of which has been reported in brain endothelial cells upon E. coli infection[31], was associated with Caspr1 in HBMECs (Supplementary Fig. 4A). Reciprocal immunoprecipitation experiments showed that endogenous Caspr1 was co-precipitated with FAK in HBMECs (Fig. 4a). The association of Caspr1 with FAK was increased in a time-dependent manner in HBMECs infected with wild-type E. coli, E44 (Fig. 4b). Interestingly, in HBMECs infected with ZD1 (ibeA deletion strain), the Caspr1-FAK association was significantly attenuated (Fig. 4b). Tyrosine phosphorylation on Tyr397 of FAK has been shown to be involved in E. coli invasion of HBMECs[31]. Here, we found the phosphorylation of FAK at Tyr397 was increased in HBMECs infected with E44, and this increase was significantly attenuated in HBMECs infected with ZD1 strain lacking ibeA (Fig. 4c). Then recombinant IbeA protein was used to stimulate HBMECs and the immunoprecipitation results showed that IbeA protein could recapitulate the enhanced Caspr1-FAK association in HBMECs infected with E. coli (Fig. 4d). Also, IbeA protein was able to induce phosphorylation of FAK at Tyr397 in HBMECs (Fig. 4e). The increased phosphorylation of FAK at Tyr397 in HBMECs in response to E. coli infection or IbeA treatment were significantly attenuated when the Caspr1 expression in HBMECs was depleted by CRISPR-Cas9-mediated knockout (Fig. 4f, g). To further analyze whether the cytoplasmic tail of Caspr1 is required for mediating intracellular signaling, stable HBMEC cell lines transfected with Caspr1 mutant lacking the 80-amino acid cytoplasmic tail (Caspr1ΔC) were established (Supplementary Fig. 4B) and the plasma membrane localization of Caspr1ΔC were observed (Supplementary Fig. 4C). We found deletion of the cytoplasmic tail in Caspr1 prevented the increased FAK phosphorylation in HBMECs infected with E. coli (Fig. 4h). In contrast, the increased phosphorylation of Src kinase and cPLA2, which has been shown to be associated with E. coli infection[32, 33], was not affected by Caspr1ΔC mutant (Supplementary Fig. 4D). Caspr1ΔC expression in HBMECs exhibited decreased invasion of E. coli (Fig. 4i), without affecting E. coli adhesion (Supplementary Fig. 4E). These results indicated that the interaction between bacterial IbeA and

---

**Fig. 2** Caspr1 is required for E. coli penetration through the BBB. **a** Gene-targeting strategy for the generation of Caspr1 conditional knockout mice with Cre-loxP recombination system. Two loxP sites were inserted before exon 4 and after exon 1 of Caspr1 gene, respectively. Then the Caspr1[loxP/loxP] mice were crossed with VE-cadherin-Cre mice (VE-cadherin[Cre/+]), producing Caspr1[loxP/loxP]; VE-cadherin[Cre/+] mice in which Caspr1 was knocked out in endothelium (Caspr1 eKO). **b** The brain slices from Caspr1 eKO mice were prepared and stained with Caspr1 (Red) and CD31 (Green) antibodies. DAPI was used to identify nuclei (blue), with Caspr1[loxP/loxP] mice served as control. Scale, 50 μm. Images are representative of three independent experiments. **c, d** E. coli (1 × 10⁴ CFUs) was injected subcutaneously into the mice. At 18 h post-infection, **c** the blood samples were collected for bacteremia measurement and the CSF were collected and cultured to indicate the passage of bacteria through the BBB. **d** Histological examination of the brain sections by hematoxylin and eosin (HE) staining was performed to assess the inflammation of the cerebral meninges. The zoom-in view showing the meningeal thickening and neutrophils infiltration (arrows) were provided. Scale, 100 μm. The rate of meningitis occurrence was calculated by dividing the number of mice with both positive CSF culture and meningeal inflammation by the total number of mice. *$P < 0.05$ (Chi-square). **e** The mice were subcutaneously injected with E. coli ($2 \times 10^2$ CFUs/mouse). The daily mortality was recorded and survival curves were plotted. $n = 14$ (each group). $P < 0.05$, Log-rank (Mantel–Cox) test. **f** Stable transfection with shRNA targeting Caspr1 was used to knockdown Caspr1 in HBMECs. Knockdown effects were analyzed by western blot, with GAPDH as loading control. Full blots are shown in Supplementary Fig. 8. Then the HBMECs were subjected to bacterial adhesion (top panel) and invasion (bottom panel) assay. Data are presented as relative invasion compared to normal HBMECs control, defined as 100%. Values are mean ± SD of four independent experiments done in triplicate, $n = 4$. ***$P < 0.001$, one-way ANOVA. The absolute value of the control (HBMECs) in bacterial invasion assay was $3.00 \pm 1.12 \times 10^3$ CFU/well. The absolute value of adhered E. coli with normal HBMECs was $3.20 \pm 1.20 \times 10^6$ CFU/well. **g** Stable HBMECs cell line with silenced Caspr1 was transfected with full-length Caspr1 for rescue experiments. Caspr1 levels were examined by western blot (lower panel). Full blots are shown in Supplementary Fig. 8. Then the cells were used for bacterial invasion assay (upper panel). Values are mean ± SD, $n = 3$. ***$P < 0.001$, one-way ANOVA. The absolute value of the control (HBMEC) was $2.90 \pm 1.52 \times 10^3$ CFU/well. **h** HBMECs were pre-treated with the indicated concentrations of Caspr1 antibody for 30 min. Then bacterial adhesion (upper panel) and invasion (lower panel) assays were performed. Values are mean ± SD, $n = 3$. *$P < 0.05$. **$P < 0.01$, one-way ANOVA. The absolute value of the vehicle control in bacterial invasion assay was $2.90 \pm 1.26 \times 10^3$ CFU/well

host Caspr1 upon *E. coli* infection induced the recruitment of FAK to Caspr1, resulting in the activation of FAK signaling in brain endothelial cells.

It has been shown that Rho GTPase Rac1 is activated in HBMECs upon *E. coli* infection[34]. By analyzing the active form of

Rac1, GTP-bound Rac1, we found Rac1 activation in HBMECs stimulated with *E. coli* was prevented by Caspr1ΔC mutant (Fig. 4j), suggesting Rac1 might be associated with intracellular signaling of Caspr1. Moreover, FAK knockdown effectively attenuated Rac1 activation in HBMECs stimulated with *E. coli*

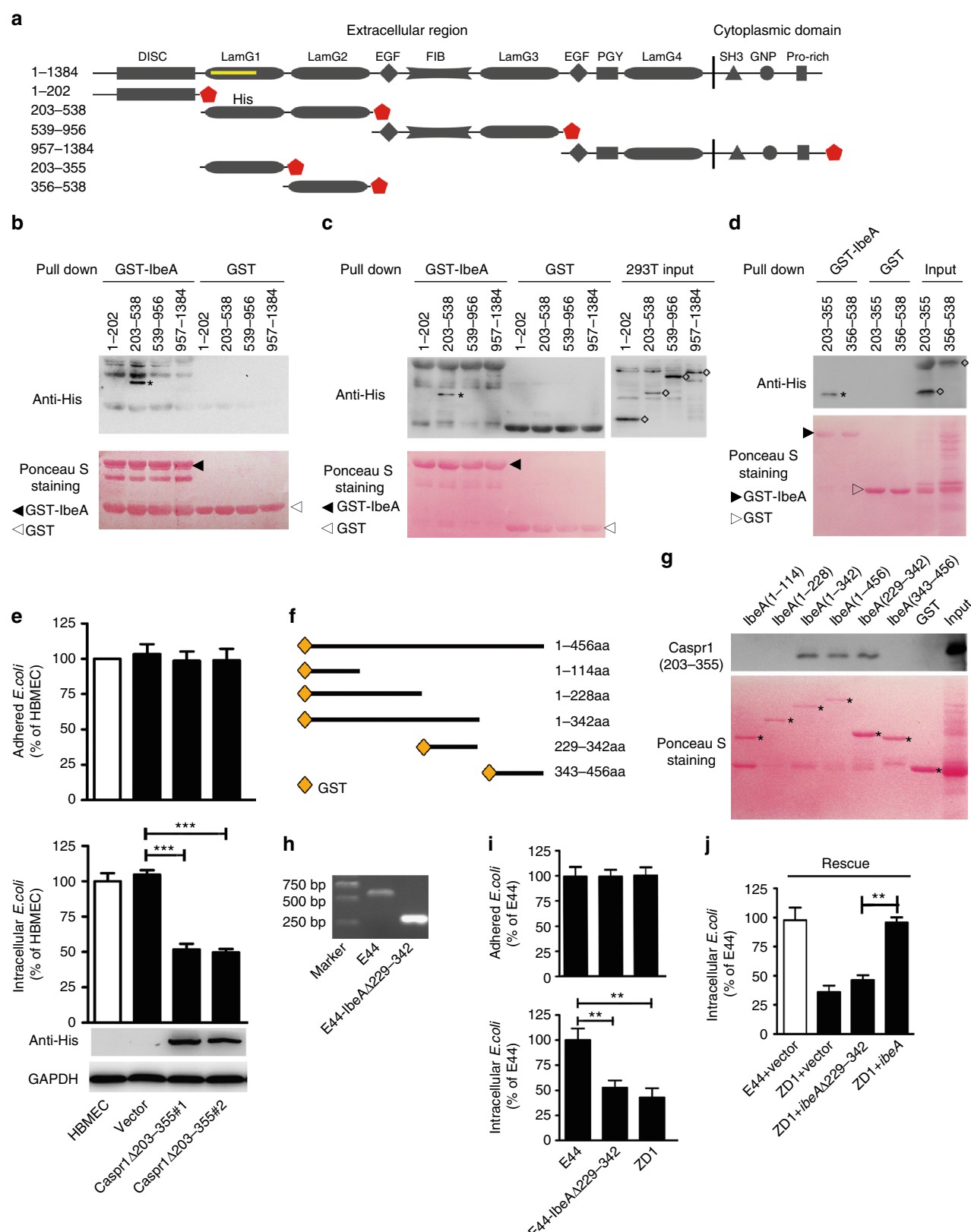

                                                                      

or recombinant IbeA protein (Fig. 4k). These data revealed that Rac1 was downstream of FAK signaling in HBMECs in response to IbeA–Caspr1 interaction.

**Caspr1(203–355) peptide can prevent *E. coli* crossing BBB in vivo**. Then we asked whether Caspr1(203–355) peptide could counteract the virulent IbeA during *E. coli* infection given that bacterial IbeA was secreted into extracellular environment upon contact with HBMECs. GST-tagged Caspr1(203–355) peptide was purified from Sf9 insect cells using a baculovirus eukaryotic expression system (Supplementary Fig. 5A). The ELISA-based protein binding assays revealed a dose-dependent increase in the binding of GST-tagged Caspr1(203–355) with IbeA (Fig. 5a). Bacterial invasion assay in the presence of purified Caspr1 (203–355) peptide indicated that Caspr1(203–355) peptide significantly reduced *E. coli* invasion into HBMECs in a dose-dependent manner (Fig. 5b).

Next, we tested whether recombinant Caspr1(203–355) peptide was able to inhibit *E. coli* crossing through the BBB in experimental meningitis. The availability of Caspr1(203–355) peptide in the blood of neonatal rats after subcutaneous injection was measured and the results showed that the half-life of Caspr1(203–355) peptide was $17.5 \pm 1.1$ h (Fig. 5c), which is suitable for in vivo blocking experiments. The neonatal rats were injected subcutaneously with $1 \times 10^5$ *E. coli* and Caspr1(203–355) peptide (30 μg/rat). At 18 h post-infection, the blood were collected to evaluate bacteremia and the results showed that bacteremia was not affected by Caspr1(203–355) peptide (Fig. 5d). Then the CSF specimens were collected to assess the bacterial penetration through the BBB, and histological examination of the brain sections were performed to analyze the meningeal inflammation. The meningeal inflammation, indicated as meningeal thickening and infiltration of neutrophils, were consistently found in the rats with positive CSF culture, but not in the rats with negative CSF culture. Our results showed that the rate of positive CSF with meningeal inflammation in rats injected with Caspr1(203–355) peptide ($n = 25$) reduced to 24% compared to the 56% in recipients of GST control ($n = 25$) ($P < 0.05$, $\chi^2$ test, Fig. 5d, e). Furthermore, we found injection of Caspr1(203–355) peptide resulted in prolonged survival of neonatal rats infected with certain dose of *E. coli* ($P < 0.05$, Fig. 5f). These results

demonstrated that administration of Caspr1(203–355) peptide was sufficient to reduce the occurrence of *E. coli* meningitis by preventing the penetration of bacteria through the BBB.

***E. coli* can invade into neurons requiring IbeA–Caspr1 interaction**. In the neonatal rats with *E. coli* meningitis, we found the presence of bacteria outside the vessels in brain's hippocampus ($4.0 \pm 1.2$ bacteria/slice, Fig. 6a), suggesting that *E. coli* could penetrate into the brain parenchyma through the BBB. This prompted us to test whether *E. coli* was able to invade neurons during infection. Primary cultured rat hippocampal neurons were infected with wild-type *E. coli* followed by transmission electron microscopy (TEM) analysis. We found the presence of *E. coli* within membrane enclosed compartment localized in the cytoplasm of neurons (Fig. 6b, five additional representative TEM images in the top panels of Supplementary Fig. 6A). The quantification revealed that the percentage of bacteria-containing neurons is $5.57 \pm 0.86\%$, and the number of bacteria within the neurons were $2.00 \pm 1.48$ in neurons infected with wild-type *E. coli* (Fig. 6c). Interestingly, when the neurons were infected with *ibeA*-deficient ZD1 strain, the percentage of bacteria-containing neurons was significantly reduced to $0.19 \pm 0.11\%$ ($P < 0.0001$, Student's *t*-test, Fig. 6c, five additional representative TEM images in the bottom panels of Supplementary Fig. 6A). Consistently, bacterial invasion assay with the cultured hippocampal neurons showed that *E. coli* were internalized into the neurons in a dose-dependent manner and the bacterial invasion was severely compromised in ZD1 strain (Fig. 6d). With a multiplicity of infection (MOI) of 100, the intracellular colony-forming units (CFUs) recovered from the hippocampal neurons infected with ZD1 were $66.5 \pm 47$, in contrast to $2260.0 \pm 384.7$ in neurons infected with wild-type E44 ($P < 0.001$, Fig. 6d). These results demonstrated that *E. coli* can invade into neurons during infection, dependent on bacterial IbeA.

Then we investigated the involvement of neuronal Caspr1 in *E. coli* invasion into neurons. The primary hippocampal neurons derived from *Caspr1*^loxP/loxP^ transgenic mice were transfected with Cre-expressing vector to ablate neuronal Caspr1 expression. The immunostaining and western blot results showed that Caspr1 were significantly reduced in the Cre-expressed neurons (Fig. 6e, bottom panel of Fig. 6f). The *E. coli* invasion assays with Caspr1-

---

**Fig. 3** Mapping the interacting domains in host Caspr1 and *E. coli* IbeA. **a** Schematic representation of the domain structure of full-length Caspr1 and its truncated forms. The coordinates of prey clones were labeled as yellow line. **b** Truncated forms of Caspr1 with His-tag were subjected to in vitro transcription and translation, and then co-incubated with the beads prebound with GST-IbeA, with GST as control, for GST pulldown assay. Precipitates were analyzed with anti-His antibody. Asterisk represents precipitated Caspr1(203–538). Images are representative of three independent experiments. Full blots are shown in Supplementary Fig. 8. **c** HEK293T cells were transiently transfected with truncated forms of Caspr1 with His-tag. The cell lysate were prepared and co-incubated with the beads prebound with GST-IbeA, with GST as control, for GST pulldown assay. Asterisk represents precipitated Caspr1 (203–538). Images are representative of three independent experiments. Full blots are shown in Supplementary Fig. 8. **d** The 203–355 and 356–538 domain of Caspr1 with His-tag were in vitro translated and then co-incubated with the beads prebound with GST-IbeA, with GST as control, for GST pulldown assay. Asterisk represents precipitated Caspr1(203–355) and rhombus represents the input proteins. Images are representative of three independent experiments. Full blots are shown in Supplementary Fig. 8. **e** HBMECs were stably transfected with Caspr1 mutant with deletion of aa 203–355 (Caspr1Δ203–355) identified by western blot, with empty vector as control. Full blots are shown in Supplementary Fig. 8. Then HBMECs were subjected to bacterial adhesion (top panel) and invasion (bottom panel) assays. Values are mean ± SD, $n = 3$. ***$P < 0.001$, one-way ANOVA. **f** Schematic illustration of the full-length IbeA protein and the truncated forms of IbeA tagged with GST. **g** In vitro translated His-tagged Caspr1(203–355) protein was subjected to GST pulldown in the presence of truncated forms of IbeA containing GST-tag, with GST as a control. Then the precipitates were analyzed by western blot with anti-His antibody. Asterisks represent the truncated forms of IbeA protein. Images are representative of three independent experiments. Full blots are shown in Supplementary Fig. 8. **h** Identification of *E. coli* mutant strain with deletion of *ibeA(229–342)* (E44-IbeAΔ229–342) by PCR amplification. The parent strain E44 served as a control. Images are representative of three independent experiments. **i** Bacterial adhesion (upper panel) and invasion (lower panel) assays were performed using E44 strain, *ibeA* deletion mutant strain ZD1 and E44-IbeAΔ229–342. Values are means ± SD, $n = 3$. **$P < 0.01$, one-way ANOVA. **j** Bacterial invasion assays were performed with ZD1 strain transformed with constructs encoding full-length IbeA or IbeAΔ229–342. E44 transformed with empty vector served as control. Values are means ± SD, $n = 3$. **$P < 0.01$, one-way ANOVA

                               

suppressed neurons showed that *E. coli* invasion was significantly decreased compared to control ($P < 0.01$, top panel of Fig. 6f). Furthermore, the transgenic mice carrying a targeted deletion of the 203–355 domain of Caspr1 were generated (Supplementary Fig. 6B, C), and then the hippocampal neurons isolated from

heterozygous mice with Caspr1(203–355) deletion ($Caspr1^{\Delta203-355/+}$) were used for *E. coli* invasion assay. The results showed that *E. coli* invasion in neurons from $Caspr1^{\Delta203-355/+}$ was reduced compared to wild-type control ($P < 0.001$, Fig. 6g). Moreover, we downregulated the expression of neuronal Caspr1

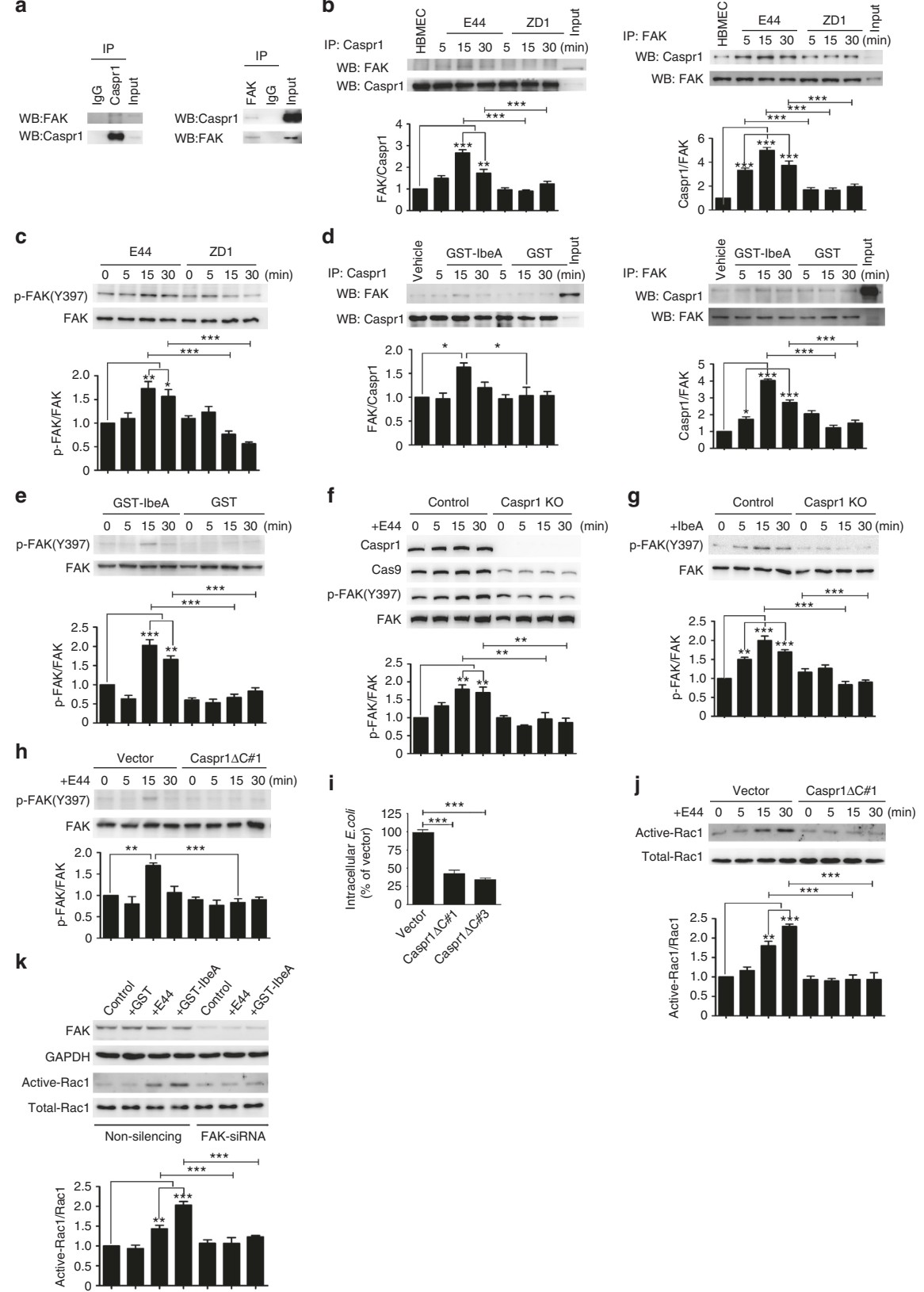

in primary rat hippocampal neurons by CRISPR-Cas9 system, and then *E. coli* invasion assay were conducted. The results showed that Caspr1 downregulation resulted in reduced *E. coli* invasion into the neurons ($P < 0.05$, Supplementary Fig. 6D). Consistently, pre-incubation with Caspr1(203–355) peptides during bacterial infection significantly reduced *E. coli* invasion into neurons (Fig. 6h). These results collectively demonstrated that neuronal Caspr1 is required for *E. coli* invasion into the hippocampal neurons.

Our additional results showed that the penetrated *E. coli* in rat hippocampus during meningitis were associated with apoptotic neurons (Supplementary Fig. 7A). Neuronal apoptosis was also observed in the cultured hippocampal neurons infected with *E. coli*, which was effectively attenuated by the inhibitors of mitochondria, suggesting the neuronal apoptosis occurred through mitochondrial apoptotic pathway (Supplementary Fig. 7B–D). The *E. coli*-induced neuronal apoptosis was significantly attenuated by pre-treatment with Caspr1(203–355) peptides as well as the deletion of *ibeA* in *E. coli* (Supplementary Fig. 7E, F). From these findings, we concluded that *E. coli* is able to penetrate the BBB and subsequently invade into hippocampal neurons causing apoptosis, which is dependent on the interaction between bacterial IbeA and neuronal Caspr1.

## Discussion

Interactions between bacterial ligands and host cell receptors are the critical steps in the pathogenesis of bacterial infection. Identification of novel host cell receptors is likely to elucidate the pathogenesis and also provide targets for the prevention and/or treatment of bacterial infection. In this study, we identified Caspr1 as the host receptor for bacterial IbeA in *E. coli* meningitis. Caspr1(203–355) peptide is sufficient to counteract IbeA to prevent *E. coli* meningitis. Our data highlight Caspr1 as a potential target for intervention of *E. coli* meningitis.

Meningitis and urinary tract infection represent common extraintestinal *E. coli* infection. IbeA is shown to be unique to *E. coli* isolates from CSF of infected neonates[35] and is present in meningitis-causing *E. coli* strains, but not in the urosepsis *E. coli* isolates[36]. The *ibeA* gene, containing a 1368-bp open reading frame, was initially identified as a 250-bp DNA fragment encoding an 8.2 kDa protein, named as *ibe10*[10]. Using this 8.2 kDa protein fragment as a bait, a binding protein of Ibe10, named as IbeA10R was identified, which is a surface protein shared 75% homology to serum albumin[29]. However, a subsequent study demonstrated that the previously identified *ibe10* is actually part of the full-length *ibeA* gene encoding 50 kDa IbeA protein[9]. Thus, whether IbeA interacted with Ibe10R remained to be determined in this new context. Another reported IbeA binding partner is vimentin[37], an intermediate filament protein. The binding of IbeA with vimentin was limited to in vitro biochemical assays[38, 39], and the contribution of this binding to *E. coli* meningitis remains unresolved.

In this study, we identified host transmembrane protein Caspr1 as the interacting partner of bacterial IbeA. Caspr1 is specifically expressed in brain microvessels, but not in the vessels of non-brain organs (e.g., liver, kidney, and lung). We found Caspr1, primarily localized at the luminal side of brain endothelium, is required for the penetration of circulating *E. coli* through the BBB and development of meningitis. Genetic ablation of Caspr1 in the endothelium inhibited passage of *E. coli* through the BBB, which is in line with previous studies that *ibeA* deletion in *E. coli* reduced its ability to invade brain endothelial cells and penetrate into the brain[9, 10]. We further found that binding of IbeA with Caspr1 triggered intracellular FAK and Rac1 signaling, which was activated in brain endothelial cells upon *E. coli* infection[31, 34]. These findings established that endothelial Caspr1 acts as a receptor of bacterial IbeA essential for penetration of *E. coli* through the BBB.

Protein sequence comparison revealed that Caspr1 did not exhibit any homology with vimentin. We considered vimentin may act as a downstream binding molecule of IbeA after bacterial entry into brain endothelial cells via IbeA–Caspr1 interaction, and the reasons for this consideration include: (1) the transmembrane localization of Caspr1 make it physically accessible to *E. coli* IbeA before bacterial internalization. In contrast, the cytoplasmic localization of vimentin indicates that its binding with IbeA can occur only after bacteria penetration into the cytoplasm and vimentin was recruited to the membrane lipid raft upon activation[38], which was at the late stage of bacterial infection; (2) In HBMECs infected with *E. coli*, the activation of Caspr1, revealed by FAK recruitment and phosphorylation, was peaked at 15 min after infection (Fig. 4b, c), which is earlier than the activation of vimentin (30 min)[38]; (3) The intracellular signaling associated with vimentin is likely to be in the downstream of Caspr1, i.e., Caspr1 promoted FAK activation (Fig. 4a–h), whereas vimentin activation led to ERK/NF-κB nuclear signaling[38, 39] which has been well demonstrated as a downstream pathway of FAK[40, 41].

IbeA bound with the extracellular N-terminal lam-G domain in Caspr1. The lam-G domain is present in a number of basement membrane constituent proteins including laminin, agrin,

---

**Fig. 4** Intracellular FAK-Rac1 signaling is activated upon IbeA–Caspr1 interaction. **a** HBMECs lysates were immunoprecipitated (IP) with Caspr1 and FAK antibody, respectively, and then the precipitated proteins were analyzed by western blot (WB) using antibodies against FAK and Caspr1. **b** HBMECs were infected with E44 or ZD1 (with MOI of 100) for the indicated times and the cell lysates were subjected to IP analysis as described in (**a**). **c** HBMECs were infected with E44 or ZD1 for the indicated times, and the cell lysates were examined by western blot with antibodies against FAK and phosphorylated FAK at Tyr397. **d** HBMECs were treated with GST-tagged IbeA protein or GST alone for the indicated times, and then the cells were lysed and subjected to IP analysis as described in (**a**). **e** HBMECs treated with GST-IbeA or GST were lysed and the samples were subjected to western blot to analyze FAK phosphorylation. **f, g** The Caspr1 in HBMECs was knocked out by CRISPR-Cas9 technique verified by western blot. The FAK phosphorylation in Caspr1-depleted HBMECs in response to *E. coli* infection (**f**) and GST-IbeA treatment (**g**) was examined by western blot. **h** HBMECs were stably transfected with Caspr1 mutant with deletion of cytoplasmic tail (Caspr1ΔC), then the HBMECs were infected with wild-type *E. coli* (E44) for the indicated times. FAK phosphorylation was analyzed by western blot. HBMECs transfected with vector served as control. **i** HBMECs stably transfected with Caspr1ΔC were subjected to bacterial invasion assay. Values are means ± SD from three independent experiments. ***$P < 0.001$, one-way ANOVA. **j** HBMECs stably transfected with Caspr1ΔC were infected with E44 for the indicated times, and then the cells were lysed and the activation of Rac1 was measured. **k** HBMECs were transiently transfected with FAK siRNA, with non-silencing siRNA as a control. Then the cells were treated with E44 or GST-IbeA for 30 min and Rac1 activation was examined. FAK levels were analyzed by western blot with GAPDH as a loading control. All the images are representative of three independent experiments. The full blots are shown in Supplementary Fig. 8. The densities of the bands in western blots were quantified and analyzed for statistical significance. *$P < 0.05$; **$P < 0.01$; ***$P < 0.001$, one-way ANOVA

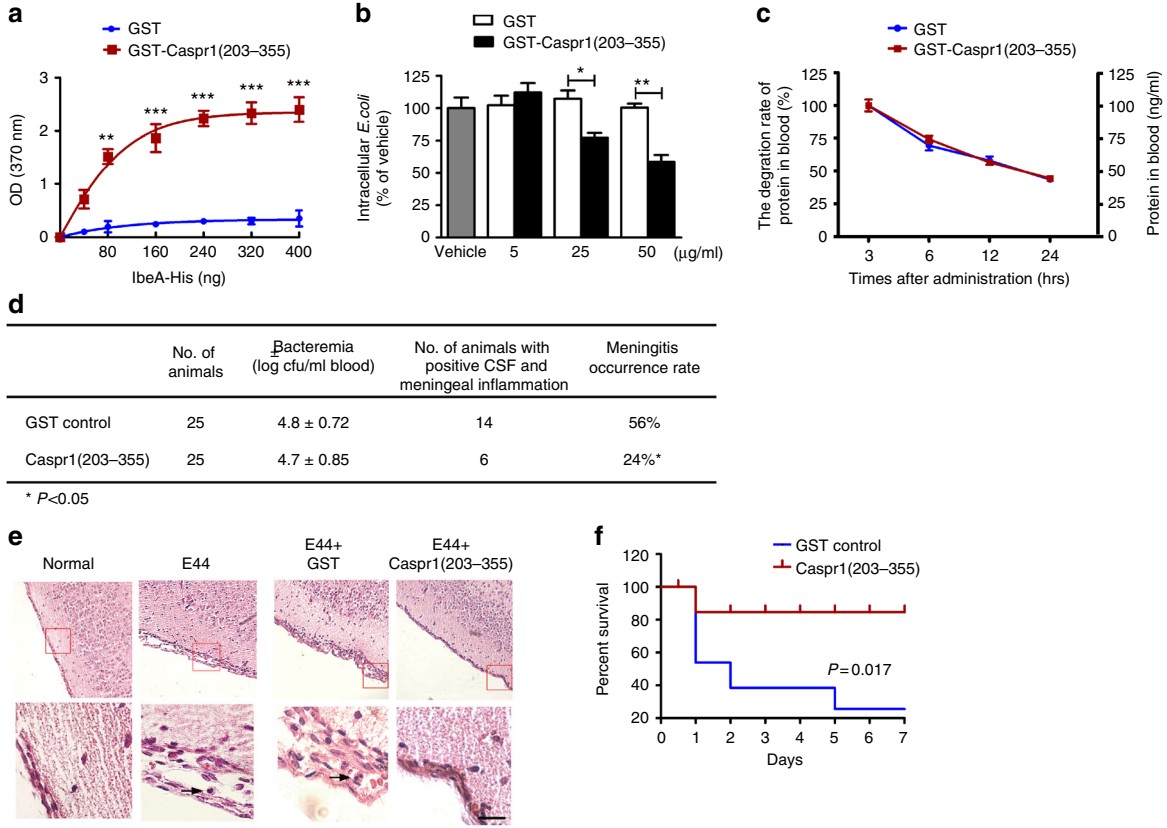

**Fig. 5** Caspr1(203–355) peptides attenuates bacterial penetration through the BBB leading to reduced *E. coli* meningitis in neonatal rats. **a** Recombinant His-tagged IbeA proteins were added to the plate coated with GST-tagged Caspr1(203–355) protein, with GST coated plate as control. The binding was assessed by in vitro binding assay by measuring the absorbance at 370 nm. Values are mean ± SD from three independent experiments. **$P < 0.01$; ***$P < 0.001$, one-way ANOVA. **b** *E. coli* were pre-incubated with indicated concentrations of Caspr1(203–355) recombinant protein for 30 min and then bacterial invasion assays were performed. Values are mean ± SD from five independent experiments. *$P < 0.05$, **$P < 0.01$, one-way ANOVA. **c** Half-life of Caspr1(203–355) recombinant protein in rat blood (30 µg/rat). Values are mean ± SD from three independent experiments. **d**, **e** The neonatal rats were subcutaneously inoculated with $1 \times 10^5$ CFUs of E44 pre-treated with GST-Caspr1(203–355) peptide (30 µg/rat), with GST protein as control. After 18 h, **d** the blood samples were collected for bacteremia measurement. The CSF were collected and cultured for bacteria. **e** The brains were harvested and sectioned for HE staining, and the representative images were provided. The bottom panels are a higher magnification of the boxed regions in the top panels. Arrows indicate neutrophils. Scale, 100 µm. The rate of meningitis occurrence was calculated by dividing the number of rats with both positive CSF and meningeal inflammation by the total number of rats. *$P < 0.05$, Chi-square test. **f** Neonatal rats were subcutaneously injected with $4 \times 10^2$ CFUs of E44 strain together with GST-Caspr1(203–355) peptide (30 µg/rat), with GST protein as control. The daily mortality was recorded and survival curves were plotted. $n = 14$ (each group). $P < 0.05$, Log-rank (Mantel–Cox) test

perlecan, and collagen[42]. Several ligands from pathogens bind with laminin and collagen during infection of mammalian cells. Owing to the extracellular localization of laminin and collagen, their roles in microbial infection are primarily limited to regulating the adhesion of microbes to mammalian cells[43]. For example, Hlp protein of *Mycobacterium leprae* binds with α2-laminin and promotes the mycobacterial adherence to Schwann cells[44, 45]. In our study, Caspr1 was functionally distinct from laminin although its extracellular portion contains lam-G domains homology to those in laminin proteins. Endothelial Caspr1 is specifically associated with *E. coli* invasion into brain endothelial cells, without affecting bacterial adhesion. Consistently, as the interacting partner of Caspr1, bacterial IbeA is not involved in bacterial adhesion to brain endothelial cells as a surface-anchored adhesin because IbeA was not localized at bacterial membrane (Supplementary Fig. 1A, B), but rather associated with *E. coli* invasion[9, 10]. Thus IbeA–Caspr1 interaction participated in the bacterial entry step instead of adhesion.

Several intracellular molecules in brain endothelial cells were found to be activated by *E. coli* infection, such as FAK, PI3K/Akt,

Src, cPLA2, and Rho GTPases[31–33], however, the upstream receptors that lead to their activation remain unclear. Here we found transmembrane receptor Caspr1 was physically associated with FAK, and IbeA–Caspr1 interaction enhanced Caspr1-FAK association and promoted FAK activation. FAK was upstream of PI3K/Akt in HBMECs infected with *E. coli*, because PI3K/Akt activation was abolished by dominant-negative mutants of FAK[31]. These suggested that IbeA–Caspr1 interaction on the membrane surface of brain endothelial cells recruits FAK to Caspr1, leading to activation of FAK/PI3K signaling. Our study thus identified Caspr1 as an upstream receptor to elicit intracellular FAK/PI3K signaling in brain endothelial cells upon *E. coli* infection. In contrast, the *E. coli*-induced phosphorylation of Src kinase and cPLA2, which has been shown associated with *E. coli* invasion[32, 33], was not affected by Caspr1ΔC mutant. Given that both *ibeA* deletion in *E. coli* and Caspr1ΔC mutant expression in brain endothelial cells caused 50–60% reduction, but not complete abolition, in *E. coli* invasion, it is likely that there are additional yet undefined factors in bacteria and host cells that mediate the activation of Src and cPLA2 signaling.

Infection of neurons by microbes has been reported in a variety of viruses, but the invasion of neurons by bacteria is much less common. To our knowledge, *Listeria monocytogenes* is one of the few, if not the only, bacteria that can invade into neurons[46, 47]. Here, we found *E. coli* was able to invade into hippocampal neurons, largely dependent on the bacterial IbeA as well as neuronal Caspr1. We noticed that *E. coli* can invade into neurons

with moderate efficiency, the efficiency of which is similar to that in HBMECs (Supplementary Fig. 6E), which is in line with the neurotropic property of meningitic *E. coli*. Our findings presented a novel concept that meningitis-causing *E. coli* exploit the same strategy, IbeA–Caspr1 interaction, to infect distinct hosts including brain endothelium and then hippocampal neurons. The neuronal apoptosis in bacterial meningitis was usually attributed

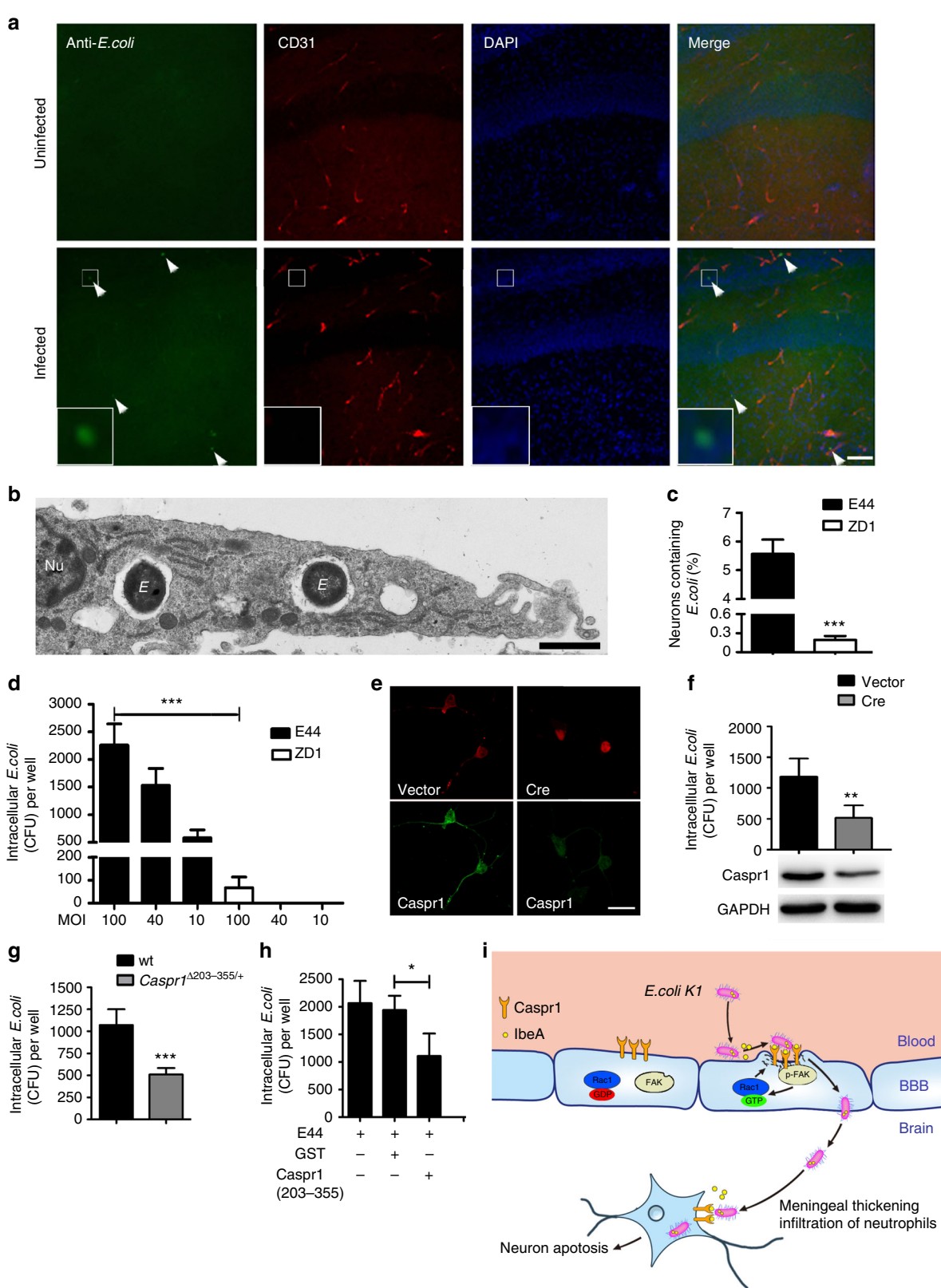

to bacteria-induced inflammatory responses[48]. Here we found the invasion of *E. coli* into hippocampal neurons directly induced apoptotic cell death (Supplementary Fig. 7). These findings revealed an alternative pathway to produce neuronal apoptosis by the internalized pathogenic *E. coli* during meningitis.

Bacterial meningitis continues to be an important cause of mortality and morbidity despite advances in antimicrobial chemotherapy that act by inhibiting bacterial viability[49]. Alternative strategies targeting virulence factors of microbes are emerging as attractive directions for therapy of infectious disease[50]. We found administration of Caspr1(203–355) peptide reduced the occurrence of *E. coli* meningitis and prolonged the survival of neonatal mice, suggesting Caspr1(203–355) peptide can counteract virulence factor IbeA and attenuate IbeA–Caspr1 interaction critical for bacterial passage through the BBB in *E. coli* meningitis. Thus Caspr1(203–355) peptide is an inhibitor of IbeA–Caspr1-mediated *E. coli* infection.

We noticed that a small portion of the rats injected with Caspr1(203–355) still developed to meningitis. We found the meningeal thickness, the levels of PMN numbers and glucose concentration in CSF were at similar levels in the rats injected with Caspr1(203–355) peptides that developed to meningitis ($n = 6$), compared to the rats with meningitis in control group injected with GST alone ($n = 14$) (Supplementary Fig. 5B–D). These results suggested that when *E. coli* meningitis occurred, the subcutaneous injection with Caspr1(203–355) peptides may have little effect on the degree of meningitis. Thus, subcutaneous injection of Caspr1(203–355) peptides specifically act on the initial entry step of bacteria into the CNS, i.e, the penetration of bacteria through the BBB, to reduce the occurrence of *E. coli* meningitis.

Our TEM results revealed that Caspr1 was not only localized at the luminal side of brain microvessels, but was also in the cytoplasm of brain endothelial cells. Consistently, the immunostaining identified localization of Caspr1 not only at the plasma membrane but also in the perinuclear region of HBMECs. The perinuclear localization of Caspr1 likely indicated the presence of Caspr1 in endoplasmic reticulum (ER) because Caspr1, bearing mannose-rich N-glycans, was reported to be transported from ER to plasma membrane[51, 52].

Caspr1 contains a large extracellular region consisting of a factor VII/discoidin region, 4 laminin G domains, a fibrinogen domain and an epidermal growth factor domain, suggesting its role in intercellular communications. Neuronal Caspr1 is a major component of the septate junctions formed between axons and paranodal loops suggesting its role in the signaling between axons and glial cells[26]. To our knowledge, there is no report about the function of endothelial Caspr1 in brain endothelium under physiological condition. We did not observe obvious phenotype in Caspr1 eKO mice, which survive to adulthood without any evident defects. The physiological function of Caspr1 in brain endothelium is an interesting topic that requires further investigations.

In summary, we identify mammalian Caspr1 as a novel receptor of *E. coli* IbeA. The interaction between bacterial IbeA and host Caspr1 is critical for *E. coli* penetration through the BBB and invasion into neurons (see the proposed model in Fig. 6i). Blocking IbeA–Caspr1 interaction attenuats entry of *E. coli* into the CNS resulting in reduced meningitis occurrence, and protects neurons from apoptosis. We conclude that Caspr1 is a host receptor of bacterial IbeA and could be a potential target for prevention and therapy of *E. coli* meningitis.

## Methods

**Antibodies and reagents**. Anti-Caspr1 (ab34151), anti-Caspr1 (ab133634), anti-calpain (ab28258), anti-VE-cadherin (ab33168), anti-fibrillarin (ab4566), anti-Caspr3 (ab89720), and anti-*E.coli* (ab20856) were purchased from Abcam. Anti-His (60001-1-Ig) was from Protein Tech Group, Inc. Anti-Caspr1 (sc-11174) and anti-GAPDH were from Santa Cruz. Anti-GST (3368), anti-FAK (3285), anti-p-FAK(Y397) (3283) were from Cell Signaling Technology. Anti-CD31 (550330) was from BD Biosciences. Anti-FAK (05-537) was from EMD Millipore. DAPI was from Roche. Secondary antibodies used for immunofluorescence and western blot were from Jackson Immuno Research Laboratories.

**Bacterial strains**. *E. coli* strain RS218 (O18:K1:H7) with K1 capsule is a clinical isolate from the CSF of a newborn infant with meningitis[11], and E44 is a spontaneous rifampin-resistant mutant of RS218. ZD1 is an *ibeA* in-frame-deletion mutant of E44[9]. E44 or ZD1 was grown overnight at 37 °C in brain heart infusion broth (BD Biosciences) with rifampin (100 μg/ml).

**Cell culture**. HBMECs were a generous gift from Dr. K. S. Kim (Johns Hopkins University, Baltimore, MD). The HBMECs were immortalized by transfecting the cells with construct containing the SV40-large T antigen. The immortalized HBMECs were morphologically and functionally similar to the primary HBMECs, which expressed FVIII-Rag, Ulex europus agglutinin I as well as carbonic anhydrase IV, and were able to take up acetylated low-density lipoprotein[53]. HBMECs were cultured in RPMI 1640 medium, with 10% fetal bovine serum (FBS, Hyclone), 10% Nu-serum (BD Biosciences), 2 mM glutamine, 1 mM sodium pyruvate, 1 × non-essential amino acid and 1 × MEM vitamin. Primary culture of hippocampal

**Fig. 6** *E. coli* could be internalized into neurons, dependent on IbeA–Caspr1 interaction. **a** Neonatal rats were subcutaneously injected with $1 \times 10^5$ CFUs of *E. coli*. After 18 h, the brains of rats with positive CSF were sectioned and stained with *E. coli* (Green) and CD31 (Red). DAPI was used to identify nuclei (blue). Arrows indicated *E. coli* in the brain parenchyma (4 ± 1.2 bacteria/slice, 27 slices from 9 rats from three independent experiments). Scale, 100 μm. Images are representative of three independent experiments. **b** Primary cultured rat hippocampal neurons were infected with wild-type *E. coli* (E44), and then the cells were subjected to transmission electron microscopy (TEM) analysis. Images are representative of three independent experiments. E: *E. coli*, Nu: nuclear. Scale bar: 1 μm. **c** Primary cultured rat hippocampal neurons were infected with E44 and *ibeA*-deleted ZD1 strain, respectively. Then the cells were subjected to quantitative TEM analysis. The percentage of neurons with internalized *E. coli* were calculated as neurons containing bacteria divided by total neurons in the field of vision. Values are mean ± SD from three independent experiments. ***$P < 0.001$, Student's *t*-test. **d** Bacterial invasion assay was performed using primary rat hippocampal neurons ($2 \times 10^5$ cells/well) infected with E44 or ZD1, with various MOI as indicated. Values are mean ± SD from three independent experiments. ***$P < 0.001$, Student's *t*-test. **e, f** Primary hippocampal neurons derived from *Caspr1^{loxP/loxP}* mice were transfected with Cre-expressing vector to ablate Caspr1 expression in neurons, with empty vector as control. The expression of Caspr1 was examined by immunostaining (**e**) and western blot (bottom panel, **f**). Full blots are shown in Supplementary Fig. 8. Images are representative of three independent experiments. Scale, 100 μm. Then *E. coli* invasion assay with Caspr1-suppressed neurons were performed (top panel, **f**). Values are mean ± SD from three independent experiments. **$P < 0.01$, Student's *t*-test. **g** Transgenic mice carrying a targeted deletion of the 203–355 domain of Caspr1 were generated, and then the neurons isolated from heterozygous Caspr1^{203-355/+} mice were used for *E. coli* invasion assay, with neurons from wild-type mice as control. Values are mean ± SD from three independent experiments. ***$P < 0.001$, Student's *t*-test. **h** Cultured rat hippocampal neurons were infected with E44 pre-incubated with Caspr1(203–355) peptide and then bacterial invasion assay was performed, with GST protein as control. Values are mean ± SD from three independent experiments. *$P < 0.05$, Student's *t*-test. **i** Proposed model. Bacterial IbeA is released upon contact with brain endothelium, and interacts with the luminal Caspr1 to activate intracellular FAK/Racl signaling. Then the *E. coli* are internalized into brain endothelial cells and penetrate through the BBB leading to meningeal inflammation. The penetrated *E. coli* in the brain parenchyma is able to invade neurons producing neuronal apoptosis, which is also dependent on IbeA–Caspr1 interaction

neurons were performed as described previously[54]. The cells were incubated at 37 °C in a 5% $CO_2$, 95% air humidified atmosphere.

**Transgenic mice**. All the animal experiments were performed according to Guidelines for Animal Care in China Medical University (CMU) and were approved by the CMU Animal Care and Use Committee. $Caspr1^{loxP/loxP}$ mice with C57BL/6 background were generated using gene-targeting techniques in C57BL/6 N mouse embryonic stem cells (Cyagen Biosciences Inc.). $Caspr1^{loxP/loxP}$ mice were crossed with VE-cadherin-Cre mice ($VE$-$cadherin^{Cre/+}$) in which expression of Cre-recombinase is under the control of the endothelial-specific $VE$-$cadherin$ promoter[30], yielding $Caspr1^{loxP/+}$; $VE$-$cadherin^{Cre/+}$ mice. Then $Caspr1^{loxP/+}$; $VE$-$cadherin^{Cre/+}$ mice were bred with $Caspr1^{loxP/loxP}$ to produce $Caspr1^{loxP/loxP}$; $VE$-$cadherin^{Cre/+}$ mice in which $Caspr1$ was knocked out in endothelium (Caspr1 eKO). The Cre-negative $Caspr1^{loxP/loxP}$ littermates were used as control. The homozygous offspring of Caspr1 eKO are viable with similar body weight compared to wild-type mice and survive to adulthood without any evident defects.

The $Caspr1^{\Delta203-355/+}$ transgenic mice were generated using gene-targeting techniques in mouse embryonic stem cells. Two loxP sites were inserted between exon 5 and exon 8 of $Caspr1$ gene for deletion of the 203–355 domain of Caspr1, without affecting the reading frame of $Caspr1$ gene. $Caspr1^{\Delta203-355/+}$ mice were self-crossed to produce $Caspr1^{+/+}$ and $Caspr1^{\Delta203-355/+}$ mice. For unrecognized mechanisms, the viable homozygous $Caspr1^{\Delta203-355/\Delta203-355}$ mice are very few.

**Yeast two-hybrid screening**. The Matchmaker Library Construction & Screening Kit (Clontech) was used for screening according to manufacturer's instructions. Briefly, full-length $E. coli$ $ibeA$ gene was cloned into pGBK as bait, and transformed into yeast strain Y187 (Clontech). The bait strain was mated with strain AH109 (Clontech) containing HBMECs cDNA library cloned into pGAD, and the yeast diploid were selected on SD/-Ade-His-Leu-Trp (-AHLT) medium containing X-α-Gal. The positive clones were obtained and identified by DNA sequencing. AH109 co-transformed with SV40-large T antigen and p53 was served as a positive control. AH109 co-transformed with SV40-large T antigen and Lamin C was used as a negative control.

**Bacterial invasion assay**. $E. coli$ invasion into HBMECs was assessed as described[10, 55]. Briefly, bacteria were added to confluent HBMECs cultured in 24-well plate with multiplicity of infection (MOI) of 100. The cells were incubated at 37 °C for 1.5 h to allow invasion to occur. The intracellular bacteria were determined after the extracellular bacteria were killed by gentamicin (100 μg/ml). The cells were then thoroughly washed and lysed with 0.5% Triton X-100. The released intracellular bacteria were enumerated as colony-forming unit (CFU) by plating on LB agar plates. For blocking studies, HBMECs were pre-incubated with Caspr1 antibody (10, 30, 60 μg/ml, respectively) for 30 min before addition of bacteria. HBMECs treated with the isotype IgG were used as control. When indicated, $E. coli$ was pre-incubated with Caspr1(203–355) recombinant peptide for 1 h on ice followed by bacterial invasion assay, with GST as control.

For invasion of neurons by $E. coli$, the primary cultured hippocampal neurons were incubated with bacteria at 37 °C for 1 h. The number of intracellular bacteria was determined after the extracellular bacteria were killed by incubation with gentamicin (100 μg/ml) for 1 h at 37 °C. The cells were thoroughly washed and lysed, and the released intracellular bacteria were enumerated as CFU by plating on LB agar plates.

**Bacterial adhesion assay**. The $E. coli$ were added to confluent HBMECs cultured in 24-well plate with multiplicity of infection (MOI) of 100. The plates were incubated in the incubator (37 °C, 5% $CO_2$, 95% humidity) for 1.5 h to allow adhesion to occur. Then the cells were washed three times with RPMI 1640 media to remove unbound bacteria, and were lysed with 0.5% Triton X-100. The lysates were plated on LB agar plates and cultured overnight to count the bacterial colony-forming unit (CFU) for quantifications.

**Plasmid construction and transfection**. Human $Caspr1$ cDNA was obtained by RT-PCR from HBMECs. Then the truncated forms of $Caspr1$ were amplified for subcloning. The primers used for amplification and the cloning vectors were listed in Supplementary Table 2. The constructs (described in Supplementary Table 2) were transfected into cells using Lipofectamine 2000 (Invitrogen) according to manufacturer's instructions.

**Stable HBMEC cell line with Caspr1 knocked out**. The single-guide RNA (sgRNA) targeting $Caspr1$ gene was designed and synthesized by Obio Technology Corporation (Shanghai, China). The $Caspr1$ sgRNA (CTGTATG-CACGCTCCCTGGG) was cloned into pLenti-U6-CMV-EGFP vector to obtain pLenti-U6-Caspr1-gRNA-CMV-EGFP construct. The empty vector was used as a control. HBMECs were cultured in 35-mm dish and transfected with lentivirus (MOI = 20:1) expressing Cas9 (pLenti-CMV-Puro-P2A-3Flag-spCas9, Obio Techology Corp.). After 24-h incubation, puromycin (1 μg/ml) was added to select stable transfected cells. HBMECs stably expressing Cas9 were further transfected with lentivirus containing pLenti-U6-Caspr1-gRNA-CMV-EGFP. The cells were

digested with trypsin solution 24 h after transfection and seeded in 96-well using limited dilution method to obtain monoclonal cells. Western blot was used to verify the knockout of Caspr1 in HBMECs.

**RNA interference**. Small interfering RNA (siRNA) sequences targeting human $Caspr1$ corresponding to the coding region, TGAGCATGATGGACGCTGCTA (nucleotides 1647 to 1667) and CAGTTCCTTTGTTCGTGACTA (nucleotides 3300 to 3320), were generated, respectively. The non-silencing siRNA sequence (TTCTCCGAACGTGTCACGT) was used as a control[55]. The targeting sequences were synthesized and cloned into pRNA-U6.1/Neo (GenScript). The recombinant constructs with short hairpin RNA (shRNA) were stably transfected into HBMECs by Lipofectamine 2000 (Invitrogen).

For transient RNA interference, siRNA targeting $Caspr1$ (nucleotides 2459 to 2477, GGGUCUUCCUAGAGAAUAAUU), $Caspr3$ (nucleotides 3365 to 3383, GCACAAAGAAACAAGUCAUU), and $FAK$ (nucleotides 2161 to 2179, CCCAGGUUUACUGAACUUAUU) were obtained from Genepharma Corp. and transiently transfected into HBMECs using Lipofectamine 2000, respectively. The non-silencing siRNA (UUCUCCGAACGUGUCACGUU) was used as a control.

**Cell viability assay**. The Premixed WST-1 Kit (Beyotime, China) was used to evaluate the viability of HBMECs. Briefly, $1 \times 10^3$ cells were seeded in the 96-well plates. At the indicated times, the culture medium was removed and WST-1 solution were added to each well, followed by incubation at 37 °C for 1 h. The plates were then read at 450 nm with a microplate reader.

**Cell apoptosis assay**. For TUNEL staining, In Situ Cell Death Detection Kit (Roche) was used to label the apoptotic cells according to manufacturer's instructions. After staining, the brain sections were mounted and analyzed with confocal microscope.

To detect neuronal apoptosis, primary cultured rat hippocampal neurons were incubated with $E. coli$, and then Annexin V-FITC Apoptosis Detection Kit (Sigma) was used to assess the expression of Annexin V and PI with flow cytometer. When indicated, $E. coli$ was pre-treated with Caspr1(203–355) recombinant proteins, with GST as a control.

**Real-time reverse transcription (RT)-PCR**. The total RNA isolated with TRIzol reagent (Invitrogen) was reverse transcribed using M-MLV reverse transcriptase (Promega). Real-time PCR was performed on an ABI 7500 real-time PCR system (Applied BioSystems) with a SYBR premix Ex $Taq$ kit (Takara Biotechnology) according to the manufacturer's instructions. The primers for Caspr family members are listed in Supplementary Table 3. The amplification conditions were as follows: 95 °C for 30 s and 40 cycles of 95 °C for 5 s and 60 °C for 34 s. The comparative cycle threshold (CT) method was used to calculate the relative gene expression level, with GAPDH as the internal control. Real-time PCR products were analyzed on agarose gel electrophoresis and verified by DNA sequencing.

**Isogenic in-frame deletion in $E. coli$**. Isogenic in-frame deletion in $E. coli$ was performed as described previously[11]. The coding sequence of $ibeA$ gene with aa 229–342 deletion was obtained by overlap PCR and was cloned into suicide vector pCVD442 using $Eco$RI and $Sal$I. The obtained construct was introduced into E44 from SM10λpir by plate mating with selection for rifampin and ampicillin resistance. The resulting colonies were picked and grown to the late logarithmic phase in LB broth without selection. Then serial dilutions were spread on LB agar plates containing 5% sucrose without NaCl. Sucrose-resistant colonies were tested for loss of ampicillin resistance, which was indicative of the loss of suicide vector sequence. PCR amplification was performed to identify the gene deletion, with forward primer (CCATTTATTAAGCGTGAA) and reverse primer (TTTAAGATGACCATTTCCA).

**Recombinant protein expression and purification**. The coding sequence of aa 203–355 in Caspr1 were subcloned into pFastBac1 and transfected into Sf9 insect cells. Then the Sf9 cells were harvested and lysed by sonication in Buffer A (50 mM Tris–HCl, 300 mM NaCl, 1% NP-40, pH 8) containing protease inhibitors (Roche). After centrifugation, the supernatant was loaded on Glutathione Sepharose 4B (GE Healthcare). The beads were eluted with Buffer B (50 mM Tris–HCl, 300 mM NaCl, 40 mM GSH, pH 8).

The truncated forms of $ibeA$ (listed in Supplementary Table 4) were cloned into pGEX4T-3 and transformed into $E. coli$ BL21 (DE3). The bacteria were grown to stationary phase, and then IPTG was added to 1 mM for additional 4 h. The cells were pelleted by centrifugation and resuspended in ice-cold PBS containing lysozyme (1 mg/ml). Cells were disrupted by sonication and Triton X-100 was added to 1%, followed by incubation at 4 °C for 30 min. Insoluble materials were removed by centrifugation and the supernatant was purified by affinity chromatography with Glutathione Sepharose 4B.

**Generation of anti-IbeA monoclonal antibody**. Four BALB/c mice were immunized by subcutaneous injection of 60 μg of purified recombinant IbeA at weeks 0, 2, 4, and 6. The immune sera of mice were collected and tested by ELISA to verify

the seroconversion. The spleens of mice whose serum had the highest titer was collected for the production of hybridomas. Myeloma SP2/0 cells were grown in DMEM supplemented with 10% fetal bovine serum. For fusion, SP2/0 cells were centrifuged, resuspended in DMEM, and combined with splenocytes at a ratio of 1:10. After centrifugation at $300 \times g$ for 10 min at room temperature, cells were resuspended in 1 ml of polyethylene glycol solution (50%, w/v; MW 1450 in Dulbecco's NaCl/Pi without calcium), and distributed in 96-well plates. After hypoxanthine/aminopterin/thymidine medium selection, the supernatants were assessed by ELISA assay. The hybridomas were cloned by limiting dilution and the produced immunoglobulins were purified on Protein A Sepharose CL-4B column.

**Western blot.** The cells were washed with ice-cold phosphate-buffered saline (PBS) and prepared with radioimmunoprecipitation assay (RIPA) buffer (50 mM Tris–HCl, 150 mM NaCl, 1% NP-40, 0.5% sodium deoxycholate, 0.1% sodium dodecyl sulfate) containing protease inhibitor cocktail. The samples were separated by SDS-PAGE and then transferred to polyvinylidene difluoride (PVDF) membrane (Millipore) by semi-dry transfer cell (Bio-Rad). The PVDF membrane was blocked with 5% nonfat milk and incubated with the primary antibody (1:1000 dilution) at 4 °C overnight. Then the blots were incubated with a horseradish peroxidase (HRP)-conjugated secondary antibody (1:8000 dilution, Santa Cruz Biotech) for 1 h at room temperature. Immunoreactive bands were visualized by Super Signal West Pico chemiluminescent substrate (Thermo Fisher Scientific Inc.) using LAS-3000 mini imaging system (Fuji Film). Each experiment was repeated at least three times. For quantification, the protein band intensities of the western blot images were quantified with ImageJ software (National Institutes of Health, Bethesda, MD). Data are represented as mean intensity of bands from three independent experiments.

**Immunoprecipitation.** Cells were washed with ice-cold PBS and lysed with lysis buffer (50 mM Tris, 150 mM NaCl, 2 mM EDTA, 2 mM EGTA, 1% Triton X-100, 1 mM sodium orthovanadate, 25 mM β-glycerophosphate, 1 mM phenylmethylsulfonyl fluoride) containing protease inhibitors. The cell lysates were centrifuged, and the supernatant was collected. The protein content was determined by the Bradford method. A total of 1 mg of protein was incubated with appropriate antibody (3 µg/ml) overnight at 4 °C and then incubated with protein A/G-agarose (Santa Cruz Biotech). The proteins precipitated from immune complexes were eluted in SDS sample buffer for western blot. Each experiment was repeated at least three times.

**Immunofluorescence.** HBMECs grown on coverslips were washed with PBS and fixed with 4% paraformaldehyde. Fixed cells were permeabilized with 0.2% Triton X-100 and then blocked with 5% bovine serum albumin (BSA) in PBS. Then the cells were stained with antibody against Caspr1 (1:100 dilution) and VE-cadherin (1:100 dilution) and then incubated with secondary antibody conjugated with Alexa488 and Alexa594 (1:200 dilution, Invitrogen), respectively. Following DAPI staining, the coverslips were mounted and analyzed under confocal laser scanning microscopy (Zeiss LSM880).

For brain slice staining, rat brains were post-fixed with 4% paraformaldehyde and then the coronal sections (100 µm) were incubated in 5% normal donkey serum for 1 h at room temperature. The slices were incubated with primary antibodies recognizing Caspr1 (1:100 dilution) and CD31 (1:100 dilution) overnight at 4 °C. E. coli was stained with FITC-conjugated E. coli-specific antibody (1:200 dilution, Abcam). Each experiment was repeated at least three times.

**Transmission electron microscopy (TEM).** The rats were anaesthetized and transcardially perfused with 4% paraformaldehyde and 0.25% glutaraldehyde in 0.1 M phosphate buffer (PB, pH 7.4) for 15 min. Then the brains were isolated and post-fixed in 4% paraformaldehyde, 0.5% glutaraldehyde and 0.06% picrate in 0.1 M PB for 4 h at 4 °C, and washed four times in 0.1 M PBS at 4 °C. Brains were dehydrated at 70%, 85%, and 95% ethanol for 20 min each time. Then the tissues were permeated with 95% alcohol and LR White resin (1:1) for 1 h, 95% alcohol and LR White (1:2) for 1 h followed by LR White for 1 h. The brains in LR White were polymerized at RT and then cut to thin slices (90–100 nm thickness). The brain slices were blocked in 5% donkey serum in PBS for 1 h and incubated with primary antibodies recognizing Caspr1 (1:100 dilution) for 2 h, and then washed and incubated with donkey anti-rabbit IgG coupled to 10-nm gold particles (1:200 dilution). Brain slices were then post-fixed in 1% glutaraldehyde for 10 min and exposed to uranyl acetate in 70% ethanol for 3 min. The slices were washed in ddH$_2$O and incubated in lead citrate solution for 3 min. Brain slices were washed and dried for imaging with transmission electron microscope (H7650, Hitachi).

For the experiments with E. coli-infected neurons, primary cultured rat hippocampal neurons were infected with E. coli for 60 min at 37 °C. Then the cells were pre-fixed for 2 h using 2.5% (v/v) glutaraldehyde in 0.1 M sodium cacodylate buffer (SCB) (pH 7.2). The cells were washed in 0.2 M SCB three times and post-fixed in 1% (w/v) osmium tetroxide in 0.2 M SCB at room temperature for 1 h, then were dehydrated through a graded series of ethanol and 100% acetone. The dehydrated cells were then infiltrated with acetone-Epon 812 resin mixtures and 100% Epon 812 resin. Ultra-thin serial sections were collected on copper formvar-coated slot grids, stained with 2% (w/v) uranyl acetate and lead citrate, and visualized under electron microscope.

**GST pulldown assay.** Equal amounts of GST and IbeA-GST fusion proteins were immobilized with Glutathione Sepharose 4B. Then different forms of Caspr1 protein derived from TNT coupled transcription/translation system (Promega), or from Caspr1 transfected 293T cells, or from HBMECs lysate, was applied and co-incubated with Glutathione Sepharose 4B prebound with GST-IbeA overnight at 4 °C, with GST as control. The bound proteins were washed with binding buffer (20 mM Tris, 50 mM NaCl, 10% glycerol, 1% Nonidet P-40) and the samples were analyzed by western blot. Each experiment was repeated at least three times.

**Rac1 activation assay.** Rac1 activation was determined by Rac1/Cdc42 activation assay kit (Millipore) following the manufacturer's instructions. Briefly, cells were treated with E. coli for indicated times and were lysed, and the lysate were incubated with PAK1 PBD coupled to Glutathione Sepharose beads. After precipitation, the beads were washed four times, eluted in SDS sample buffer and immunoblotted with antibody against Rac1. Each experiment was repeated at least three times.

**In vitro binding assay.** MaxiSorp flat-bottom plates (Nunc) were coated with GST-tagged Caspr1(203–355) protein (4 µg/well) for 1 h at 37 °C followed by an overnight incubation at 4 °C. A plate coated with GST protein was served as negative control. Then the plates were blocked with 1% BSA for 2 h at 37 °C, and His-tagged IbeA protein was applied. Plates were incubated for 2 h at 37 °C and then overnight at 4 °C. The plates were washed and incubated for 2 h at 37 °C with mouse anti-His antibody (1:8000). After washing, horse radish peroxidase-conjugated goat anti-mouse IgG (Santa Cruz Biotech) was applied. After 2 h incubation at 37 °C, the plates were washed and 3,3′,5,5′-tetramethylbenzidine (Sigma) was added. Following reaction with peroxidase, the plates were read at 370 nm with microplate reader (SpectraMax M5, Molecular Devices Company, USA).

**Neonatal rodent model of experimental E. coli meningitis.** E. coli meningitis was induced in 5-day-old C57BL/6 mice of both sexes (Beijing Vital River Laboratory Animal Technology Co.) as described[56]. Briefly, all neonatal mice were randomly divided into groups based on litter, gender and body weight and were subcutaneously injected with $1 \times 10^4$ CFU of the wild-type E. coli (E44). At 18 h after bacterial inoculation, blood and CSF specimens were obtained for quantitative cultures. Bacterial penetration across the BBB was defined as positive culture in CSF. The bacteria in the samples obtained from infected mice were determined by plating serial dilutions on brain heart infusion agar plates. The numbers of polymorphonuclear (PMN) and white blood cells in CSF were calculated by XN-1000 automatic hematology analyzer (Sysmex, Japan). The concentrations of glucose in CSF were analyzed with Glucose detection kits (Nanjing Jiancheng Bioengineering Institute, China) according to manufacturer's instructions.

For neonatal model of E. coli meningitis with rats, the procedure is similar as in mice except that 5-day-old Sprague–Dawley rats of both sexes (Beijing Vital River Laboratory Animal Technology Co.) were injected with $1 \times 10^5$ CFU as previously described[5]. When indicated, the rats were subcutaneously injected with E. coli with GST-tagged Caspr1(203–355) peptides (30 µg/rat), with GST as a control. In certain experiments, the rats were subcutaneously injected with Caspr1 antibody (40 µg/rat), with isotype IgG as a control.

**Histological analysis.** The neonatal rats or mice were killed after bacterial inoculation, and the brains were removed and immersion-fixed overnight in 4% paraformaldehyde. Coronal sections were embedded in paraffin, cut into 5-µm-thick sections, and stained with hematoxylin and eosin for histological evaluation.

**Statistical analysis.** The quantitative variables are expressed as the mean ± SD. All analyses were performed using GraphPad Prism software. Statistical significance between two groups was analyzed by Student's t-test. One-way analysis of variance (ANOVA) or two-way ANOVA was used to compare multiple groups. P value of <0.05 was considered significant.

**Data availability.** The relevant data supporting the findings of this study are available in this article and its Supplementary Information files, or from the corresponding authors upon request.

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

## Acknowledgements

This work was supported by the National Natural Science Foundation of China (30970120, 31171291, 31571057, 31771258, 81171537, 81201255, 31670845), the National 1000 Plan Program, the NIH grants (NS91102, AI113273 and AI126176) from the National Institutes of Health, the Innovation Team Program Foundation of Ministry of Education of China (IRT13101, IRT_17R107), the National Research Foundation for the Doctoral Program of Higher Education of China (20132104110019), the National Key R&D Program of China (2016YFC1302400), the Program of Distinguished Professor of Liaoning Province, the Foundation of Liaoning Province (LT2011011, LR2012025, LS201609) and the Intramural Program of China Medical University (JQ20160002).

## Author contributions

W.-D. Z. and D.-X. L. conducted most of the experiments; J.-Y.W., Z.-W.M., K. Z., Z.-K.S., X.-W.Z., Q. L., W.-G.F., X.-X.Q., D.-S.S., B. L., and Q.-C.L. performed some experiments; K. S. K. and L. C. provide suggestions on preparing manuscript; W.-D.Z. and Y.-H.C. designed the experiments and wrote the manuscript.

## Additional information

**Competing interests:** The authors declare no competing interests.

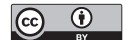

