## [Peer Review File · Nature Communications]

Reviewers' comments:

Reviewer #1 (Remarks to the Author):

This is a nice study identifying a host receptor, Caspr1, for the E. coli virulence factor IbeA to facilitate bacterial penetration of the Blood-Brain Barrier and development of E. coli meningitis. The experimentation is well executed and generally support the author's claims. However the manuscript itself requires additional information and rationale to better convey results. Specific points are below.

1. The authors should provide a better explanation of why they screened for IbeA-host interaction. They suggest that other bacterial factors were examined, but it is not clear what results were. Why was the IbeA target used? Also was caspr1 the only factor that interacted with ibeA? It would be helpful to include this data. Again the authors should provide a rationale for why they looked at IbeA.
2. Fig 1. Please discuss the normal function of caspr1 in brain endothelium. Explain it is not only in the lumen (Fig 1G)
3. Fig S1. Please explain why ibeA is not present in the WT strain, seems to be observed only when in presence of hBMEC?
4. Fig 2. A better explanation of how the KO mouse was generated would be helpful. The authors say there is more meningitis in WT mice compared to the KO mouse, however they have not presented evidence of inflammation, PMN infiltration or meningeal thickening. They should show this by looking at the entire brain before zooming into a specific section. This will give a better idea of the overall inflammatory response in all infected mice.
5. In the methods section the authors need to clarify methods for experiments with mice vs rats
6. Fig 3. No data is shown for LamG and it should be (page 7)
7. When caspr1 is knocked down in brain endothelium are there any other effects? Possibly this impacts cell viability? This should be addressed.
8. Fig. 4 please better explain how quantification is done and the meaning of the results.
9. Fig 5. The half life was looked at in mouse blood, but then actually treatment experiments were done in rats. The authors need to repeat experiments use rat blood.
10. Fig. 6. These experiments looking at neuronal apoptosis seems to come out of no where. I would suggest they might be better suited for a supplemental figure or they further develop this line of investigation. Please indicate what number of experiments were performed, "images are representative of X independent experiments"
11. Fig S5 please check these data, it seems that they may need to gate on the cell population that represents total cell death (upper right quadrant)...if to then will be no difference in apoptosis when caspr is added. Please check this and correct or explain.
12. In the discussion the authors tend to overstate their data. For example they say "blocking interaction prevented entry of E. coli..." but the data show they still observed 50% bacterial entry. A better way to discuss would be that caspr1 contributes to entry. The statement on evolution is not correct (page 17). E. coli likely did not evolved to cause meningitis and kill its host as this is a dead

end (likely an accident) for the bacteria. Again please add more discussion about what the normal function of caspr1 is and why the bacteria might be highjacking this factor

Reviewer #2 (Remarks to the Author):

The study by Zhao et al. reports on the identification of Caspr1 as a novel receptor for the E.coli K1 virulence factor IbeA which has been shown to be essential for the onset of bacterial meningitis. The authors convincingly show that IbeA binds to a specific lam-G domain of surface exposed Caspr1 which is essential for invasion of the endothelial cells. Further, the study demonstrates that this interaction activates the FAK-Rac1 signaling pathway and is involved in the invasion of neurons in neonatal rats with E. coli meningitis leading to apoptosis. These are important findings and will advance the field considerably. Hence, the study is worth publishing in Nature Communications. However, there are some major shortcomings in the presentation of the manuscript which need to be corrected. More importantly, several conclusions need to be further substantiated by additional analysis.

General:

There are many, many occasions where the article 'the' is missing or where you find mistakes in Grammar. Therefore, the text of the ms needs to be edited by a native speaker - which raises some doubts whether KSK might have seen the text.

Specifics:

line 1: Caspr1 is not a receptor for E. coli meningitis but a receptor for E. coli bacteria causing meningitis. Hence that title should be augmented.

line 80: HBMEC are an immortal cell line - this should be clarified.

line 94ff: Please explain why there is such a dominant staining for VE-cadherin in the nucleus ?

line 100 and Fig. 1G: Caspr1 is also found intracellularly- comment please. Further there should be a control for immunogold labeling as the labeling appears rather weak.

line 111: Do Caspr1 eKO mice show any phenotype ?

line 116: why are bacteria injected s.c. and not i.v. ?

line 134: In the resubmitted ms it should read 'bottom' or 'lower' panel not 'middle' panel.

line 137: 'was' should be changed to 'were'

line 141 and Fig. S2G: ko of Caspr3 could be improved

line 150: which lam-G domain is addressed ?

line 154: pull-down assays

line 156 and elsewhere: it should better read aa 203-255 or 'the 203-255 domain

line 157, line 160: comment on the additional bands please

line 167: HBMEC (Caspr1 Δ 203-255) mutant: what about endogenous Caspr1 ? Please comment !

line 168: again it should read 'bottom panel'

line 17/174: and the binding was examined...

line 194 ff and Fig. 4B,C: These evaluations need densitometry to be able to state significant attenuation

line 235 ff: Why s.c. injection of the Caspr1(203-255) peptide ?

line 261: very few bacteria (?)

line 851: Ponceau S

Fig.4E: labeling should be centered

Fig.5: Why do only about half of the animals get infected ?

Reviewer #3 (Remarks to the Author):

The authors present good evidence that Caspr1 is important for the ability of E. coli to traverse the blood brain barrier in their cell culture models and rodent models of meningitis. The best evidence for this is the fact that they have generated mice where Caspr1 can be conditionally deleted. These mice show reduced E. coli in the CSF. They have also conducted structure/function studies and have identified a peptide of Caspr1 that is able to reduce the ability of E. coli to infect. They have demonstrated the efficacy of this peptide in cell culture and rodent models.

However, in addition to these findings, the authors make a very provocative claim that E. coli can directly invade neurons and cause their apoptosis. If true, this is a significant finding. However, because of its novelty, it is important to make sure that these experiments are done rigorously- which unfortunately they are not. As detailed below, the authors will need to conduct additional experiments if they are to make the claim that E.coli directly infects neurons and induces its apoptosis. Most importantly, for the authors to conclude that the invasion of E. coli in neurons requires the neuronal Caspr1 protein, they will need to conduct the invasion experiments in neurons that are deleted for Caspr1.

1) The experiments in Fig. 4 are designed to show that *ibeA* acts via its association with Caspr1 to activate FAK signaling. While the data shown are consistent with this model, it would be ideal to have HBMECs that are depleted for Caspr1, or cells isolated from the Caspr1-deficient mice, to show no activation of FAK signaling in these cells in response to *ibeA*. The experiments with the Caspr1 C-terminal mutants are reasonable, but is unclear why the authors did not use cells that are depleted for Caspr1.

2) In Fig. 5H, have the authors looked at survival beyond the 5 day period? Also, it would be ideal to use the Caspr1 KO mice in these type of survival studies. The authors have generated these mice and it is unclear why they did not use these mice more frequently in their study.

3) In Fig. 6, the authors are presenting evidence that the exposure of E. coli to hippocampal neurons results in the invasion of E. coli into neurons that results in neuronal apoptosis. As the authors point out, the evidence that any bacteria can directly invade neurons is very limited in the literature. Thus, the data presented in this manuscript that E. coli can directly infect neurons and trigger its apoptosis needs to be evaluated carefully. The EM micrograph shown in Fig. 6B is interesting but would need to be quantitated (% of neurons showing infection with wildtype and ZD1 E. coli). The fluorescent images shown in 6A do not have the resolution to conclusively show that E. coli is inside the neurons.

4) It is important to point out that the fact that the Caspr1 peptide blocks the invasion of E.coli in neurons does not show that the neuronal Caspr1 protein is essential for the invasion of E. coli into neurons. Why? Because the Caspr1 peptide here is a reagent that binds to and inactivates the E. coli *ibeA* protein. To show that the neuronal Caspr1 protein is important, one would need to delete Caspr1 from these neurons and then evaluate the ability of E. coli to invade neurons. As the authors have already generated the Caspr1 KO mice, this can be done by utilizing the hippocampal neurons isolated from their Caspr1 KO mice.

5) It is rather unexpected that neurons would undergo apoptosis (TUNEL positivity) within 30 min of exposure to the E. coli (Fig. S5B). There may not be any precedence of neurons undergoing apoptosis so fast. Thus, if this is true, the authors would need to do more to characterize such fast kinetics of death. Is this death inhibited with a caspase inhibitor? Is the mitochondrial pathway involved? (can it be blocked with Bax deficiency, are neuronal mitochondria releasing cytochrome c).

Minor Points:

In Fig. 1D, the authors should provide more details of their nuclear marker 38F3.

In Fig. 1G, the electron micrographs do not have the clarity to show that Caspr1 is localized to the luminal side of brain microvessels.

In Fig. 2, the authors use the number of mice with "positive CSF" to quantify the percent of mice in which the bacteria enters the BBB. Is it standard practice to not include the actual numbers of bacteria in the CSF (the way the CFUs are shown for bacteria levels in the blood)? Is there a threshold below or above which the CSF is considered negative or positive, respectively, for the bacteria?

Reviewer #4 (Remarks to the Author):

This is an interesting paper and the data that Caspr1 acts a receptor for the entry of E. coli IbeA entry into HBMEC cells is pretty convincing as is the data to suggest that drugs that block this interaction have the potential to be developed into a new treatment strategy. The data included in Figs. 4 and 6 is much less convincing and appears preliminary in nature. There are numerous comments below in regards to most of the figures, many reflect the limited information provided for many of the experimental conditions and can be addressed by modifying the text/figure legends. However, additional experiments would be needed to strengthen the interpretation of experiments presented in Figs. 4 and 6.

1. Fig.1

A. In the Y2H screen were any other potential interacting proteins identified? If so, what? How many colonies were screened? Were 4 distinct clones of Caspr1 isolated? Coordinates of clones? More details of results of screen should be included in methods or supplement.

B. How are expts in 1B/C conducted? Is IbeA being expressed in the HEK and HBMEC cells? Or are cell lysates being run over GST- columns prebound with proteins of interest? This is unclear from both figure legend and text. If co-expressed, isn't it surprising that an interaction is detected between presumably cytosolic IbeA and Caspr1 _ as its interacting domain is presumably extracellular.

C. In Fig. 1C, is anything known about the titer of bacteria in CSF? Is the titer in the CSF of the eKO lower than the control mice? If this is the case, would include this data.

D. The images shown in Fig. 1E suggest that most of Caspr1 is mostly perinuclear rather than membrane localized, yet the fractionations studies suggest that it is all located in the membrane. This is mentioned, but not really emphasized. I guess this might explain why the proteins co-IP when presumably co-expressed in HEK and HBMEC cells in Fig. 1B. (Also in these images is it surprising that most VE-cadherin appears to be nuclear?)

E. In Fig. 1D, details in method regarding adherence assay are very limited. Are the infected cells extensively washed before number of bacteria quantified? Presumably yes- but never mentioned. Also, by quantifying all the bacteria present- how can they differentiate between adherent and invasive bacteria? Lastly, the Y-axis label is confusing. From methods it appears that they are quantifying bacteria in the well, thus nothing is known regarding the % of HBMEC cells with adherent bacteria. To draw such a conclusion, they would need to image infected cells. Thus, at this point appears premature to conclude that the bacteria are not impaired in adherence.

F. In Fig. 1E, as in Fig. 1D, Y-axis is not correctly labeled. Unless imaging cells cannot determine % OF HBMEC cells with intracellular bacteria. Rather the way assay is designed, the investigators are basically comparing the number of intracellular bacteria in the cells in presence and absence of Caspr1. This number reflects both the invasion and intracellular replication of bacteria, but tell us

nothing about % of infected cells.

2. Fig. 3

A. In Fig. 3A, would be helpful to include information regarding coordinates of the fragments of Caspr1 detected in Y2H plus provide information regarding the % similarity/homology shared by the 4 LamG domains.

B. Fig. 3E, did the investigators confirm that Caspr1 delta 203-355 is stably expressed and correctly localized? If so, would state that this is the case.

C. Fig. 3, is IbeA delta 229-342 secreted by the E. coli and if so, at levels equivalent to WT IbeA? This needs to be shown to definitely demonstrate that the mutant strain defect is directly due to the inability of this region of IbeA to bind to and mediate uptake of the bacteria into the HBMEC cells.

3. Fig. 4:

A. The reciprocal IPs in 4 A are not very convincing. Suggest weak interaction, if any. The immunoblots shown are not as convincing as the Coomassie gels shown in the supplement.

B. In Fig. 4B, time is presumably in minutes- would clarify. What happens if look at later time points? Does data become more convincing? What are numbers shown in each lane- presumably quantification of interacting protein pulled down in WB? Or is it some type of ratio of the interacting protein to that of the pulled-down protein? Is anything known regarding the percentage of infected cells at this time point? If it is low, then perhaps this explains weakness of detected interactions. Can the experiment be modified to promote adhesion of E. coli to the HBMEC cells? Presumably MOI of 100 is being used here? This should be clarified.

C. In Fig. 4F is it known whether the caspr1 delta C1 is correctly localized to the membrane? If not, the significance of data is unclear. The phosphorylation data in supplemental Fig. C is not very convincing.

D. Fig. 4D. This expt is overinterpreted. Presumably bacteria invasion is markedly inhibited in the caspr1 delta C lines, thus Rac1 activation could be due to any downstream event associated with bacteria invasion. Thus the conclusion on line 219 that rac1 is associated with intracellular signaling of Caspr1 is an overstatement.

E. Fig. 4I, can the E. coli invade the FAK1 depleted cells?

4. Fig. 5

A. Fig. 5C/D. Would include dose of peptide given SC in Fig. 5C. Presumably same amt as given in 5D.

B. In Figure 5, is it thought that the peptide acts to both block as well as attenuate the degree to which mice develop meningitis? Are there differences in the appearance of the tissue section or CSF of mice treated with peptide that develop meningitis vs. those that do not? Are the sections in (E) from a mouse treated with peptide that developed meningitis (defined by positive CSF cultures). Similarly, how many of the three CSF samples shown in (F,G) were obtained from mice with evidence of meningitis – that received peptide. How many mice were studied in H (include in figure legend). Does the statement on line 254 that these mice had “experimental E. coli meningitis” mean that upon death, all the mice in Fig. H had positive CSF cultures? Or does it mean that all mice infected with doses of bacteria that survival was prolonged when mice were infected with normally lethal doses of bacteria?

5. Figure 6

(A) Were only three tiled images quantified? Were they all from the same mouse? Seems that multiple images of multiple mice should be imaged before drawing any conclusions

(B) How many neurons were imaged? How often were bacteria visualized? Is invasion dependent on IbeA? How many bacteria/neuron visualized? Need to provide some type of quantification to support the significance of the presented image. At what times post-invasion was the cell shown processed for

imaging? Same conditions as in C?

(C) How many hippocampal neurons were infected in each experiment? Presumably thousands? If so seems that invasion is very inefficient. For example if 1,000 cells are infected at an MOI of 100 – that would mean that 100,000 bacteria were initially in a well – yet only 20 bacteria are isolated post-gent treatment. And, note- gent only added after 90 minutes- so the bacterial in the wells like doubled at least two times. Thus, if the bacteria do invade the hippocampal cells, they do so with very low efficiency. However, if the infected cells are dying, as suggested by the supplemental data, than the number of quantified intracellular bacteria will be difficult to determine, because if the bacteria are released by the dead cells they will be killed by the gent. In addition, gent will be able to penetrate into the dead cells and hence kill intracellular bacteria. The investigators might test whether numbers of intracellular bacteria increase in the presence of an apoptosis inhibitor.

(D) Is anything known about Capr1 expression in the hippocampal neurons? Are those the labeled cells in Fig. 2B that were not discussed?

Responses to Reviewer's comments

Reviewer #1 (Remarks to the Author):

This is a nice study identifying a host receptor, Caspr1, for the E. coli virulence factor IbeA to facilitate bacterial penetration of the Blood-Brain Barrier and development of E. coli meningitis. The experimentation is well executed and generally support the author's claims. However the manuscript itself requires additional information and rationale to better convey results. Specific points are below.

Reply: Thanks for reviewer's encouragement about our manuscript.

1. The authors should provide a better explanation of why they screened for IbeA-host interaction. They suggest that other bacterial factors were examined, but it is not clear what results were. Why was the IbeA target used? Also was caspr1 the only factor that interacted with ibeA? It would be helpful to include this data. Again the authors should provide a rationale for why they looked at IbeA.

Reply: According to reviewer's suggestion, here we would like to explain why we screened for the interacting partner of bacterial IbeA in host cells as below. Bacterial IbeA was identified as a critical determinant of meningitis-causing *E. coli* to facilitate bacterial penetration through blood-brain barrier (BBB) (Infect Immun. 1995;63:4470-5, J Infect Dis. 2001;183:1071-8). However, the exact mechanism by which IbeA exerts its effect during *E. coli*-host interaction remains unclear. We recently found that bacterial IbeA protein could be secreted from *E. coli* bodies into the extracellular environment upon contact with host cells, i.e., brain microvascular endothelial cells which is the primary component of BBB (Fig. S1A and B). These prompted us to hypothesize that the secreted IbeA may interact with membrane proteins expressed on host cell surface. Therefore, we used yeast two-hybrid (Y2H) to screen the interacting proteins of IbeA in host cells, human brain microvascular endothelial cells (HBMECs), and then we identified Caspr1 as a novel membrane-anchored binding protein of bacterial IbeA.

As reviewer suggested, we provided the original results of Y2H screening as new Table S1 (attached below) in the revised manuscript. The results of Y2H presented us a number of candidates as potential IbeA-interacting partners in host cells. Among these candidates, we realized that Caspr1 is the only transmembrane protein localized on cell surface which is in line with our initial hypothesis that IbeA may interact with membrane proteins on host cell surface. Thus, we focused on host Caspr1 in this study by performing a series of experiments to verify the interaction between IbeA and Caspr1, as well as the importance of their interaction in *E. coli* meningitis.

In addition, there might be some misunderstanding about the bacterial factors used for yeast two-hybrid. In this study, we only used IbeA (but not other bacterial factors) as bait protein to screen the interacting protein of IbeA in host cells.

Table S1 Results of yeast two-hybrid screen using IbeA as the bait protein

Clone #	Genbank accession #	Gene name
35, 66	NM_001101	Actin, beta
10	NM_004077	Citrate synthase
40, 62, 65, 72	NM_003632	Caspr1 (contactin associated protein 1)
2, 6, 20, 22, 24, 34, 50, 61	BC065494	DEAD/H (Asp-Glu-Ala-Asp/His) box polypeptide 12
19, 46, 47, 58	NM_000402	Glucose-6-phosphate dehydrogenase
33	NM_007260	Lysophospholipase II
3, 56	NM_015889	Mediator complex subunit 15
11	AY129569	Polymerase (DNA directed), delta 1, catalytic subunit
32	NM_001105570	Nudix hydrolase 19
4, 8, 15, 25, 26, 27, 38, 39, 67, 68, 69	NM_020850	RAN binding protein 10
14, 63	NM_133452	Ribonucleoprotein, PTB binding 1
29, 57	BC013878	Thimet oligopeptidase 1
41	NM_006086	Tubulin beta 3 class III
5	NM_004651	Ubiquitin specific peptidase 11

2. Fig 1. Please discuss the normal function of *caspr1* in brain endothelium. Explain it is not only in the lumen (Fig 1G)

Reply: Thanks for reviewer's suggestion. In our study, the previously uncharacterized Caspr1 expression in brain endothelium was identified, and we found Caspr1 acts as host receptor of bacterial IbeA during *E. coli* meningitis. To our knowledge, there is no report about the function of Caspr1 in brain endothelium under physiological condition. Structural analysis of Caspr1 protein showed that Caspr1 contains a large extracellular region consisting of a factor VII/discoidin region, 4 laminin G domains, a fibrinogen domain and an epidermal growth factor domain, suggesting its role in intercellular communications. Neuronal Caspr1 is a major component of the septate junctions formed between axons and paranodal loops suggesting its role in the signaling between axons and glial cells (J Cell Biol. 1997;139: 1495-1506). In Caspr1 eKO mice, we did not observe obvious phenotype and the mice could survive to adulthood without any evident defects. Thus, the physiological function of Caspr1 in brain endothelium is an interesting topic that requires further investigations. These were added to the discussion of the revised manuscript in Page 20, Line 469-478.

As reviewer pointed out, Caspr1 was not only localized at the luminal side of brain microvessels, but was also found in the cytoplasm of brain endothelial cells (Fig. 1G). Consistently, the immunostaining results identified localization of Caspr1 not only at the plasma membrane but also in the intracellular perinuclear region of HBMECs (Fig. 1E). The perinuclear localization of Caspr1 likely indicated the presence of Caspr1 in endoplasmic reticulum (ER) because Caspr1, bearing mannose-rich N-glycans, was reported to be transported from ER to plasma membrane (J Biol Chem. 2003;278:48339-47, J Cell Biol. 2000;

149:491-502). These were added to the revised manuscript in Page 20, Line 460-468.

3. *Fig S1. Please explain why ibeA is not present in the WT strain, seems to be observed only when in presence of hBMEC?*

Reply: There might be some misunderstanding about the results in Fig. S1A. In the first lane on the left of the western blot image (top), the expression of IbeA, though relatively weak, was clearly observed in the bacterial pellet of wild-type E44 strain, indicating IbeA protein is indeed present in wild-type *E. coli* strain. Note the position of IbeA bands with expected molecular weight were marked by the arrow on the right. When E44 strain were co-cultured with HBMECs, a prominent IbeA band in the culture supernatant was observed (the third lane on the left), suggesting IbeA was efficiently secreted upon contact with host cells. In contrast, IbeA is barely detected in the supernatant of E44 strain cultured alone (the fourth lane on the left), suggesting little IbeA was secreted without contact with HBMECs. These data indicated that IbeA secretion is largely dependent on the contact between bacteria and HBMECs.

To avoid any misunderstands, we provided detailed information for each lane of the western blot image in the figure legend of Fig. S1A in the revised manuscript. In addition, we adjust the brightness and contrast of the image in Fig. S1A for a better presentation.

4. *Fig 2. A better explanation of how the KO mouse was generated would be helpful.*

Reply: According to reviewer's suggestion, we provided detailed information regarding the generation of Caspr1 conditional knockout mice as follows: "Caspr1^{loxP/loxP} mice were generated using gene-targeting techniques in mouse embryonic stem cells. The two loxP sites were inserted before exon 4 and after exon 1 of Caspr1 gene, respectively. Then the Caspr1^{loxP/loxP} mice were crossed with VE cadherin-Cre mice (VE-cadherin^{Cre/+}), producing Caspr1^{loxP/loxP}; VE-cadherin^{Cre/+} mice in which Caspr1 was knocked out in endothelium (Caspr1 eKO)". These were added to the figure legend of Fig. 2A in the revised manuscript.

The authors say there is more meningitis in WT mice compared to the KO mouse, however they have not presented evidence of inflammation, PMN infiltration or meningeal thickening. They should show this by looking at the entire brain before zooming into a specific section. This will give a better idea of the overall inflammatory response in all infected mice.

Reply: According to reviewer's suggestion, we provided the representative images showing meningeal inflammation (reflected as meningeal thickening and neutrophils infiltration in meninges) of the wild-type mice with meningitis, compared to the meninges without inflammation of the endothelial-specific Caspr1 knockout mice (Caspr1 eKO) (new Fig. 2D) in the revised manuscript. Furthermore, we provided the images of the entire brain before zooming into the local meninges with inflammation in the new Fig. 2D as reviewer suggested.

Note that the meningeal inflammation was consistently observed in the mice with positive CSF culture and was absent in the mice with negative CSF culture. For better understandings,

we modified the header “No. of animals with positive CSF” to “No. of animals with positive CSF and meningeal inflammation” in the table of Fig. 2C in the revised manuscript.

5. In the methods section the authors need to clarify methods for experiments with mice vs rats

Reply: Thanks for reviewer’s kind reminder. We carefully checked our manuscript and provided detailed information to clarify the methods with mice and rats in the Experimental Procedure of the revised manuscript in Page 39-40, Line 935-952.

6. Fig 3. No data is shown for LamG and it should be (page 7)

Reply: We performed sequence alignment analysis using the translated DNA sequencing results of the positive clones interacted with IbeA in yeast two-hybrid (Y2H) assays, and the analysis showed that it is corresponded to the N-terminal laminin-globular (lam-G) domain of Caspr1. These data were presented as new Fig. S3A in the revised manuscript. The coordinates of the Y2H clones were marked on the domain architecture of Caspr1 in Fig. 3A and new Fig. S3A in the revised manuscript.

7. When caspr1 is knocked down in brain endothelium are there any other effects? Possibly this impacts cell viability? This should be addressed.

Reply: We did not observe any evident defects in brain endothelial cells when Caspr1 was knocked down. According to reviewer’s suggestion, we performed cell viability assay to evaluate the effect of Caspr1 knockdown on survival of HBMECs. The results showed that Caspr1 knockdown had no effect on viability of HBMECs compared to control. The results were provided as new Fig. S2D in the revised manuscript.

8. Fig. 4 please better explain how quantification is done and the meaning of the results.

Reply: For quantification, the protein band intensities of the Western blot images were quantified with ImageJ software (National Institutes of Health, Bethesda, MD). Data are represented as mean intensity of bands from three independent experiments. According to reviewer’s suggestion, this detailed information about the quantification methods for the western blot results in Fig. 4 were provide in the Experimental Procedure of the revised manuscript in Page 36, Line 853-856. For better presentation, we added bar graphs with statistical analysis to each western blot results in Fig. 4 in the revised manuscript. By this way, it would be easier to understand the meaning of the results in Fig. 4.

9. Fig 5. The half life was looked at in mouse blood, but then actually treatment experiments were done in rats. The authors need to repeat experiments use rat blood.

Reply: We repeated the experiments in Fig. 5C with rat blood to measure the half-life time of Caspr1(203-355) peptides as reviewer suggested. The half-life time of Caspr1(203-355) peptides in rat blood is 17.5 ± 1.1 h, at similar level to that (15.6 ± 1.2 h) in mice blood. Thus,

we updated Fig. 5C with the data derived from rat blood in the revised manuscript.

10. Fig. 6. These experiments looking at neuronal apoptosis seems to come out of no where. I would suggest they might be better suited for a supplemental figure or they further develop this line of investigation. Please indicate what number of experiments were performed, “images are representative of X independent experiments”

Reply: As reviewer suggested, we performed new lines of experiments to further clarify the importance of Caspr1 in *E. coli* invasion into neurons. These new results were summarized as follows: (1) Primary hippocampal neurons derived from Caspr1^{loxP/loxP} transgenic mice were transfected with Cre-expressing vector to ablate Caspr1 expression in neurons. The results from immunostaining and western blot showed that Caspr1 were significantly reduced in the Cre-expressed neurons (new Fig. 6E, bottom panel of new Fig. 6F). Then we performed *E. coli* invasion assay with Caspr1-suppressed neurons and found *E. coli* invasion was significantly decreased compared to control (P<0.001, top panel of Fig. 6F); (2) Transgenic mice carrying a targeted deletion of the 203-355 domain of Caspr1 were generated (new Fig. S6B and C), and then the neurons isolated from heterozygous mice with Caspr1 203-355 deletion (named as Caspr1^{Δ203-355/+}) were used for *E. coli* invasion assay. The results showed that *E. coli* invasion in neurons from Caspr1^{Δ203-355/+} was reduced compared to wild-type control (P<0.001, Fig. 6G). Note that the viable homozygous mice with Caspr1 203-355 deletion (Caspr1^{Δ203-355/Δ203-355}) were very few due to unidentified mechanisms, and the primary neuron isolated from the heterozygous Caspr1^{Δ203-355/+} mice were used in Fig. 6G; (3) Primary rat hippocampal neurons were cultured and the neuronal Caspr1 were down-regulated by CRISPR-cas9 gene editing technology, and then *E. coli* invasion assay were conducted. The results showed that Caspr1 downregulation resulted in reduced *E. coli* invasion into the neurons (P<0.05, new Fig. S6D). These data collectively demonstrated that neuronal Caspr1 is required for *E. coli* invasion into neurons.

In addition, according to reviewer’s suggestion, the repeated number of experiments were indicated in the figure legend of Fig. 6 in the revised manuscript. Also, the number of independent experiments from which representative images were obtained were indicated in the figure legend of Fig. 6.

11. Fig S5 please check these data, it seems that they may need to gate on the cell population that represents total cell death (upper right quadrant)...if to then will be no difference in apoptosis when caspr is added. Please check this and correct or explain.

Reply: There might be some misunderstandings. In Fig. S5C (changed to Fig. S7E in the revised manuscript), the primary neurons with indicated treatment were stained with Annexin-V and Propidium iodide (PI) followed by flow cytometry analysis. Then we calculated the cell population in the lower right quadrant, with positive Annexin-V and negative PI, as cells undergoing apoptosis in early stage (Fig. S7E in revised manuscript). The reason we choose the

lower right quadrant for calculations is that the early apoptotic cells have phosphatidylserine flipped to the outer leaflet of membrane where it can be bound by Annexin-V, whereas PI is unable to penetrate early apoptotic cells with intact membrane so these cells will be PI-negative. Regarding the cell population in upper right quadrant in Fig. S7E, it did not necessarily represent total cell death, but could be a mixture of cells with necrosis and late apoptosis because necrotic and late apoptotic cells are stained by both Annexin-V and PI due to permeabilized plasma membrane. One cannot distinguish these two types of cell death in this cell population. Thus, in our manuscript, to determine whether IbeA-Caspr1 interaction is associated with neuronal apoptosis during *E. coli* infection, the cell population in the lower right quadrant of Fig. S7E were used for apoptosis rate analysis. Our results showed that deletion of *ibeA* in *E. coli*, as well as pretreatment with Caspr1(203-355) peptides, both significantly attenuated *E. coli*-induced neuronal apoptosis.

For a better visualization, we marked the lower right quadrant of Fig. S7E with “Apop.” to indicate the apoptotic cells in the revised manuscript.

12. In the discussion the authors tend to overstate their data. For example they say “blocking interaction prevented entry of E. coli...” but the data show they still observed 50% bacterial entry. A better way to discuss would be that caspr1 contributes to entry.

Reply: According to reviewer’s suggestion, we replaced “prevented” with “attenuated” in the sentence as follows: “Blocking IbeA-Caspr1 interaction attenuated entry of *E. coli* into the CNS resulting in reduced meningitis occurrence, and protects neurons from apoptosis.” in the revised manuscript in Page 20, Line 482-483.

The statement on evolution is not correct (page 17). E. coli likely did not evolved to cause meningitis and kill its host as this is a dead end (likely an accident) for the bacteria.

Reply: We modified this statement as follows: “Our findings presented a novel concept that meningitis-causing *E. coli* exploit the same strategy, IbeA-Caspr1 interaction, to infect distinct hosts including brain endothelium and then hippocampal neurons.” in the revised manuscript in Page 18, Line 431-433.

Again please add more discussion about what the normal function of caspr1 is and why the bacteria might be hijacking this factor

Reply: To our knowledge, there is no report about the function of Caspr1 in brain endothelium under physiological condition. Structural analysis of Caspr1 protein showed that Caspr1 contains a large extracellular region consisting of a factor VII/discoidin region, 4 laminin G domains, a fibrinogen domain and an epidermal growth factor domain, suggesting its role in intercellular communications. Neuronal Caspr1 is a major component of the septate junctions formed between axons and paranodal loops suggesting its role in the signaling between axons and glial cells (J Cell Biol. 1997;139: 1495-1506). In Caspr1 eKO mice, we did not observe

obvious phenotype and the mice could survive to adulthood without any evident defects. Thus, the physiological function of Caspr1 in brain endothelium is an interesting topic that requires further investigations. According to reviewer's suggestion, these discussion about the normal function of Caspr1 in brain endothelium was added to the revised manuscript in Page 20, Line 469-478.

Regarding the reason why bacteria hijack Caspr1 during meningitis, our study indicated that there are 3 major reasons as follows: 1) Caspr1 is exclusively present in the brain microvessels but not in the peripheral microvessels, the location of which is appropriate for the entry of meningitis-causing *E. coli* into the brain; 2) Caspr1, with an extracellular region, is localized at the luminal side of the brain microvessels, which is readily to be exploited by the circulating bacteria when bacteremia occurred; 3) The N-terminal lamG domain of Caspr1 specifically interacted with bacterial factor IbeA, which enables the recognition by *E. coli* to subvert host cell signaling. As a result, the released IbeA derived from circulating *E. coli* is able to hijack the Caspr1 to alter the intracellular signaling pathways to enable bacterial penetration from the blood to the brain.

Reviewer #2 (Remarks to the Author):

The study by Zhao et al. reports on the identification of Caspr1 as a novel receptor for the E.coli K1 virulence factor IbeA which has been shown to be essential for the onset of bacterial meningitis. The authors convincingly show that IbeA binds to a specific lamb-G domain of surface exposed Caspr1 which is essential for invasion of the endothelial cells. Further, the study demonstrates that this interaction activates the FAK-RAc1 signaling pathway and is involved in the invasion of neurons in neonatal rats with E. coli meningitis leading to apoptosis. These are important findings and will advance the field considerably. Hence, the study is worth publishing in Nature Communications. However, there are some major shortcomings in the presentation of the manuscript which need to be corrected. More importantly, several conclusions need to be further substantiated by additional analysis.

Reply: Thanks for reviewer's support and encouragement about our manuscript.

General:

There are many, many occasions where the article 'the' is missing or where you find mistakes in Grammar. Therefore, the text of the ms needs to be edited by a native speaker - which raises some doubts whether KSK might have seen the text.

Reply: We asked an English native speaker to carefully read the whole manuscript and the grammar mistakes were corrected in the revised manuscript.

Specifics:

line 1: Caspr1 is not a receptor for E. coli meningitis but a receptor for E. coli bacteria causing

meningitis. Hence the title should be augmented.

Reply: Thanks for reviewer's helpful suggestion. We modified the title as "Caspr1 is the host receptor for *Escherichia coli* causing meningitis" in the revised manuscript.

line 80: HBMEC are an immortal cell line - this should be clarified.

Reply: We provided detailed information about HBMEC, an immortal cell line, as follows. HBMEC cell line were immortalized by transfecting the cells with construct containing the SV40-large T antigen. The immortalized HBMECs were morphologically and functionally similar to the primary HBMEC, which expressed FVIII-Rag, Ulex europus agglutinin I as well as carbonic anhydrase IV, and were able to take up acetylated low-density lipoprotein. These were added to the Experimental Procedure of the revised manuscript in Page 30, Line 717-721.

line 94ff: Please explain why there is such a dominant staining for VE-cadherin in the nucleus ?

Reply: The dominant staining of VE-cadherin in the nucleus might be non-specific staining caused by the antibody used. We repeated the experiments in Fig. 1E using new VE-cadherin antibody, and the results showed that the signal of VE-cadherin at the plasma membrane was improved whereas the signal in the nucleus was reduced (new Fig. 1E presented in the revised manuscript).

line 100 and Fig. 1G: Caspr1 is also found intracellularly- comment please. Further there should be a control for immunogold labeling as the labeling appears rather weak.

Reply: As reviewer pointed out, Caspr1 was not only localized at the luminal side of brain microvessels, but was also found in the cytoplasm of brain endothelial cells (Fig. 1G). Consistently, the immunostaining results identified localization of Caspr1 not only at the plasma membrane but also in the intracellular perinuclear region of HBMECs (Fig. 1E). The perinuclear localization of Caspr1 likely indicated the presence of Caspr1 in endoplasmic reticulum (ER) because Caspr1, bearing mannose-rich N-glycans, was reported to be transported from ER to plasma membrane (J Biol Chem. 2003;278:48339-47, J Cell Biol. 2000; 149:491-502). These were added to the revised manuscript in Page 20, Line 460-468.

Regarding the control for the image in Fig. 1G, a negative control image for immunogold staining was provided as new Fig. S1D in the revised manuscript. Furthermore, for better visualization, we adjusted the brightness and contrast of the image in Fig. 1G and a zoom-in view of the positive staining of Caspr1 indicated as gold particles at the luminal side of brain microvessels were placed on the right of Fig. 1G in the revised manuscript.

line 111: Do Caspr1 eKO mice show any phenotype ?

Reply: We did not observe obvious phenotype in Caspr1 eKO mice. The homozygous offspring with endothelial-specific Caspr1 knockout are viable with similar body weight compared to wild-type mice. These Caspr1 eKO mice survive to adulthood without any evident defects.

These were incorporated to the revised manuscript in Page 20, Line 476-477.

line 116: why are bacteria injected s.c. and not i.v. ?

Reply: Subcutaneous injection was a routinely used method to induce bacterial meningitis in neonatal rodents, which has been widely used in a series of studies (Nat Commun. 2011;2:552, J Biol Chem. 2002;277:15607-12, J Clin Invest. 1992;90:897-905). The reason that intravenous injection was not used in our study is that the tail vein of the 5-day-old neonatal mice are too small for injection.

line 134: In the resubmitted ms it should read 'bottom' or 'lower' panel not 'middle' panel.

Reply: As reviewer suggested, we corrected the “middle panel” to “bottom panel” in the revised manuscript.

line 137: 'was' should be changed to 'were'

Reply: We corrected it in the revised manuscript.

line 141 and Fig. S2G: ko of Caspr3 could be improved

Reply: We repeated the western blot experiments of Caspr3, and the image was replaced with a better one as new Fig. S2F in the revised manuscript.

line 150: which lam-G domain is addressed ?

Reply: We modified this sentence as follows: “It is therefore proposed that one of the lam-G domain in Caspr1 might be responsible for the interaction between Caspr1 and IbeA”.

line 154: pull-down assays

Reply: We corrected it in the revised manuscript.

line 156 and elsewhere: it should better read aa 203-255 or 'the 203-255 domain

Reply: We corrected all “203-538aa” to “aa 203-538” in the whole revised manuscript.

line 157, line 160: comment on the additional bands please

Reply: In the western blot images of Fig. 3B and C, the specific band of Caspr1 (aa 203-538), indicating its positive binding with GST-IbeA, was marked by asterisk according to its expected molecular weight. Yet, there are additional bands located in the left 4 lanes as well as in the lower part of western blot images in Fig. 3B and C. Because these bands showed apparently similar migration patterns between distinct lanes, we interpret these additional bands as non-specific binding.

line 167: HBMEC (Caspr1Δ203-255) mutant: what about endogeneous Caspr1 ? Please

comment !

Reply: To answer reviewer's question, we performed immunostaining experiments to examine the localization of Caspr1 Δ 203-255 in HBMECs stably-transfected with Caspr1 Δ 203-255 construct. As expected, we observed the localization of Caspr1 Δ 203-255 at the plasma membrane (new Fig. S3B in the revised manuscript). As a result, the membrane-localized Caspr1 Δ 203-255 is able to compete with the endogenous membrane-anchored Caspr1 to attenuate its function in HBMECs in response to *E. coli* virulent factor IbeA. Therefore, the *E. coli* invasion into HBMECs expressing Caspr1 Δ 203-255 was reduced (bottom panel, Fig. 3E) due to the attenuated IbeA-Caspr1 interaction.

line 168: again it should read 'bottom panel'

Reply: As reviewer suggested, we corrected the "middle panel" to "bottom panel" in the revised manuscript.

line 17/174: and the binding was examined...

Reply: In line 173, the text "and examining the binding of IbeA" was corrected to "and the binding was examined" according to reviewer's suggestion.

line 194 ff and Fig. 4B,C: These evaluations need densitometry to be able to state significant attenuation

Reply: According to reviewer's suggestion, we performed densitometric quantification of all the western blot results in Fig. 4 and the statistical significance were labeled in Fig. 4 B, C, D, E, F, G, H, J and K in the revised manuscript.

line 235 ff: Why s.c. injection of the Caspr1(203-255) peptide ?

Reply: The 203-255 domain of Caspr1 was identified that physically interacted with bacterial IbeA in our study (Fig. 3B, C, D and G). Given that bacterial IbeA could be secreted from *E. coli* when contact with brain endothelial cells (Fig. S1A), we attempt to use Caspr1(203-255) peptide as a strategy to neutralize the secreted IbeA in the peripheral blood during development of *E. coli* meningitis. Therefore, the Caspr1(203-255) peptide were injected subcutaneously into neonatal mice with experimental *E. coli* meningitis to neutralize the secreted IbeA. The results showed that injection of Caspr1(203-255) peptide reduced the occurrence rate of meningitis ($p < 0.05$, Fig. 5D and E).

line 261: very few bacteria (?)

Reply: We carefully counted the bacteria in the brain parenchyma located outside of microvessels in neonatal rats with *E. coli* meningitis, and the results showed that there were 4.0 ± 1.2 bacteria per brain slice ($n=27$ slices, from 9 rats). Though the number of bacteria is relative low, the physical existence of bacteria in brain parenchyma reliably indicated that the

circulating *E. coli* could infiltrate into the brain parenchyma through microvessels during meningitis.

line 851: Ponceau S

Reply: Thanks for reviewer's reminder, we corrected "ponceau S" to "Ponceau S" as suggested.

Fig.4E: labeling should be centered

Reply: The labeled text was slightly moved to be aligned with the center of the lanes of western blot images in the revised Fig. 4E.

Fig.5: Why do only about half of the animals get infected ?

Reply: The rate of animals with meningitis, indicated as positive bacterial culture of cerebral spinal fluid (CSF) and meningeal inflammation identified by hematoxylin and eosin (HE) staining, was primarily dependent on the number of bacteria inoculated. In our study, 1×10^5 CFU of *E. coli* were injected to the neonatal rats, resulting in a meningitis rate reached to 56 % in the control group. Thus, only in this way, whether the Caspr1(203-355) peptide treatment could increase or reduce the occurrence of meningitis will be determined by comparison with the 56 % meningitis rate in control group. Note that the experimental bacterial meningitis using neonatal rodents to induce the meningitis rate to 40-60% was a routinely used method in previous studies (Infect Immun. 2002;70:5865-9, Microb Pathog. 2004;37:287-93).

Reviewer #3 (Remarks to the Author):

The authors present good evidence that Caspr1 is important for the ability of E. coli to traverse the blood brain barrier in their cell culture models and rodent models of meningitis. The best evidence for this is the fact that they have generated mice where Caspr1 can be conditionally deleted. These mice show reduced E. coli in the CSF. They have also conducted structure/function studies and have identified a peptide of Caspr1 that is able to reduce the ability of E. coli to infect. They have demonstrated the efficacy of this peptide in cell culture and rodent models.

Reply: Thanks for reviewer's encouragement about our manuscript.

However, in addition to these findings, the authors make a very provocative claim that E. coli can directly invade neurons and cause their apoptosis. If true, this is a significant finding. However, because of its novelty, it is important to make sure that these experiments are done rigorously- which unfortunately they are not. As detailed below, the authors will need to conduct additional experiments if they are to make the claim that E.coli directly infects neurons and induces its apoptosis. Most importantly, for the authors to conclude that the invasion of E. coli in neurons requires the neuronal Caspr1 protein, they will need to conduct the invasion

experiments in neurons that are deleted for Caspr1.

Reply: According to reviewer's suggestion, we performed three sets of experiments to verify that neuronal Caspr1 is required for *E. coli* invasion into neurons. The results were summarized as follows: (1) Primary hippocampal neurons derived from Caspr1^{loxP/loxP} transgenic mice were transfected with Cre-expressing vector to ablate Caspr1 expression in neurons. The results from immunostaining and western blot showed that Caspr1 were significantly reduced in the Cre-expressed neurons (new Fig. 6E, bottom panel of new Fig. 6F). Then we performed *E. coli* invasion assay with Caspr1-suppressed neurons and found *E. coli* invasion was significantly decreased compared to control (P<0.001, top panel of Fig. 6F); (2) Transgenic mice carrying a targeted deletion of the 203-355 domain of Caspr1 were generated (new Fig. S6B and C), and then the neurons isolated from heterozygous mice with Caspr1 203-355 deletion (named as Caspr1^{Δ203-355/+}) were used for *E. coli* invasion assay. The results showed that *E. coli* invasion in neurons from Caspr1^{Δ203-355/+} mice was reduced compared to wild-type control (P<0.001, Fig. 6G). Note that the viable homozygous mice with Caspr1 203-355 deletion (Caspr1^{Δ203-355/Δ203-355}) were very few due to unidentified mechanisms, and the primary neuron isolated from the heterozygous Caspr1^{Δ203-355/+} mice were used in Fig. 6G; (3) Primary rat hippocampal neurons were cultured and the neuronal Caspr1 were down-regulated by CRISPR-cas9 gene editing technology, and then *E. coli* invasion assay were conducted. The results showed that Caspr1 downregulation resulted in reduced *E. coli* invasion into the neurons (P<0.05, new Fig. S6D). These data collectively demonstrated that neuronal Caspr1 is required for *E. coli* invasion into neurons.

1) The experiments in Fig. 4 are designed to show that ibeA acts via its association with Caspr1 to activate FAK signaling. While the data shown are consistent with this model, it would be ideal to have HBMECs that are depleted for Caspr1, or cells isolated from the Caspr1-deficient mice, to show no activation of FAK signaling in these cells in response to ibeA. The experiments with the Caspr1 C-terminal mutants are reasonable, but is unclear why the authors did not use cells that are depleted for Caspr1.

Reply: As reviewer suggested, we established new HBMEC cell lines with stably depletion of Caspr1 by CRISPR-Cas9 technique. Then we analyzed the FAK phosphorylation in Caspr1-depleted HBMECs in response to *E. coli* infection or IbeA protein. The results showed that FAK activation (measured by phosphorylation of Tyr397 by western blot) was alleviated in Caspr1-deficient HBMECs infected with wild-type *E. coli* (new Fig. 4F). Further results showed that FAK activation was reduced in Caspr1-deficient HBMECs in response to recombinant IbeA protein (new Fig. 4G). We hope these data could alleviate reviewer's concern. Note that here the Caspr1-depleted HBMECs were constructed using CRISPR-cas9 system, because it is quite challenging for us to obtain high yield of primary brain endothelial cells from Caspr1 knockout mice that is sufficient for *E. coli* invasion assay.

The methods to establish stable Caspr1-deficient HBMEC cell lines with CRISPR-cas9

techniques was provided in the Experimental Procedure in the revised manuscript.

2) In Fig. 5H, have the authors looked at survival beyond the 5 day period? Also, it would be ideal to use the *Caspr1* KO mice in these type of survival studies. The authors have generated these mice and it is unclear why they did not use these mice more frequently in their study.

Reply: We checked the original data and found that all the neonatal rats survived longer than 5 days were still alive until 1 week. According to reviewer's suggestion, we replotted the survival curves up to 1 week as new Fig. 5F in the revised manuscript. In our experience, the neonatal rats that survived longer than 5 days will remain alive for months.

Furthermore, as reviewer suggested, we repeated the studies of Fig. 5F using endothelial-specific *Caspr1* knockout mice (*Caspr1* eKO) and the survival curves were presented as new Fig. 2E. The results showed the survival time of neonatal *Caspr1* eKO mice inoculated with *E. coli* were significantly prolonged ($p < 0.05$, Fig. 2E).

Here, we would like to explain why we did not use *Caspr1* eKO mice in Fig. 5 as reviewer questioned. The major point of Fig. 5 is to clarify whether *Caspr1*(203-355) peptides was effective to prevent *E. coli* penetration through the BBB and the development of *E. coli* meningitis. Therefore, rodents with wild-type genetic background, but not *Caspr1* knockout, are appropriate experimental models to evaluate the effect of *Caspr1*(203-355) peptides on *E. coli* meningitis.

3) In Fig. 6, the authors are presenting evidence that the exposure of *E. coli* to hippocampal neurons results in the invasion of *E. coli* into neurons that results in neuronal apoptosis. As the authors point out, the evidence that any bacteria can directly invade neurons is very limited in the literature. Thus, the data presented in this manuscript that *E. coli* can directly infect neurons and trigger its apoptosis needs to be evaluated carefully. The EM micrograph shown in Fig. 6B is interesting but would need to be quantitated (% of neurons showing infection with wildtype and ZD1 *E. coli*). The fluorescent images shown in 6A do not have the resolution to conclusively show that *E. coli* is inside the neurons.

Reply: According to reviewer's suggestion, the electron microscopy (EM) results in Fig. 6B were quantified and the results showed that the percentage of bacteria-containing neurons is 5.57 ± 0.86 % when neurons were infected with wild-type *E. coli*. As reviewer suggested, we performed further EM experiments with *ibeA*-deleted ZD1 strain, and we found the percentage of bacteria-containing neurons infected with ZD1 strain was significantly reduced to 0.19 ± 0.11 % compared to wild-type *E. coli* ($P < 0.0001$, new Fig. 6C). For each group, i.e., neurons infected with wild-type *E. coli* as well as neurons infected with ZD1, we provided 5 additional representative EM images in new Fig. S6A in the revised manuscript. These data indicated that *IbeA* is essential for *E. coli* invasion into neurons.

Regarding the fluorescent images in Fig. 6A, there might be some misunderstandings. From Fig. 6A, we observed the bacteria in the brain hippocampus outside the vessels in neonatal rats

with *E. coli* meningitis, which suggested that *E. coli* could penetrate into the brain parenchyma through the BBB during meningitis. As reviewer mentioned, the resolution of confocal images of brain slices is unable to conclude that *E. coli* is inside the neurons. Thus, we performed EM analysis to determine whether *E. coli* can invade into neurons. As shown in Fig. 6B and C, as well as new Fig. S6A, the EM results consistently revealed the presence of bacteria in primary cultured hippocampal neurons infected with *E. coli*, demonstrating that *E. coli* can successfully invade into neurons.

4) It is important to point out that the fact that the Caspr1 peptide blocks the invasion of E.coli in neurons does not show that the neuronal Caspr1 protein is essential for the invasion of E. coli into neurons. Why? Because the Caspr1 peptide here is a reagent that binds to and inactivates the E. coli ibeA protein. To show that the neuronal Caspr1 protein is important, one would need to delete Caspr1 from these neurons and then evaluate the ability of E. coli to invade neurons. As the authors have already generated the Caspr1 KO mice, this can be done by utilizing the hippocampal neurons isolated from their Caspr1 KO mice.

Reply: Thanks for reviewer's great suggestion. In the revised manuscript, we performed three sets of experiments to verify that Caspr1 is essential for *E. coli* invasion into neurons. These new results were summarized as follows: (1) As reviewer suggested, primary hippocampal neurons were isolated from Caspr1^{loxP/loxP} transgenic mice, and then transfected with Cre-expressing vector to ablate Caspr1 expression in neurons. The results from immunostaining and western blot showed that Caspr1 were significantly reduced in the Cre-expressed neurons (new Fig. 6E, bottom panel of new Fig. 6F). Then we performed *E. coli* invasion assay with Caspr1-suppressed neurons and found *E. coli* invasion was significantly decreased compared to control (P<0.001, top panel of Fig. 6F); (2) Transgenic mice carrying a targeted deletion of the 203-355 domain of Caspr1 were generated (new Fig. S6B and C), and then the neurons isolated from heterozygous mice with Caspr1 203-355 deletion (named as Caspr1^{Δ203-355/+}) were used for *E. coli* invasion assay. The results showed that *E. coli* invasion in neurons from Caspr1^{Δ203-355/+} mice was reduced compared to wild-type control (P<0.001, Fig. 6G). Note that the viable homozygous mice with Caspr1 203-355 deletion (Caspr1^{Δ203-355/Δ203-355}) were very few due to unidentified mechanisms, and the primary neuron isolated from the heterozygous Caspr1^{Δ203-355/+} mice were used in Fig. 6G; (3) Primary rat hippocampal neurons were cultured and the neuronal Caspr1 were down-regulated by CRISPR-cas9 gene editing technology, and then *E. coli* invasion assay were conducted. The results showed that Caspr1 downregulation resulted in reduced *E. coli* invasion into the neurons (P<0.05, new Fig. S6D). These data collectively demonstrated that neuronal Caspr1 is required for *E. coli* invasion into neurons.

5) It is rather unexpected that neurons would undergo apoptosis (TUNEL positivity) within 30 min of exposure to the E. coli (Fig. S5B). There may not be any precedence of neurons undergoing apoptosis so fast. Thus, if this is true, the authors would need to do more to

characterize such fast kinetics of death. Is this death inhibited with a caspase inhibitor? Is the mitochondrial pathway involved? (can it be blocked with Bax deficiency, are neuronal mitochondria releasing cytochrome c).

Reply: Rapid apoptosis of mammalian cells in response to pathogen infection has been reported in previous studies. For example, 15.3 % of primary white blood cells from mice underwent apoptosis after a 30 min exposure to *Bordetella bronchiseptica* (*Cell Microbiol.* 2006;8:758-68). Co-incubation of Caco-2 intestinal epithelial cells with *E. coli* for 1 h showed significant increase in apoptotic cell death (0.59 %) compared with control (0.02 %) (*Toxicol In Vitro.* 2006;20:1435-45). The vesicular stomatitis virus-induced apoptosis occurred within 1 h in cervical cancer HeLa cells treated with recombinant tumor necrosis factor (*FEBS Lett.* 1998;426:179-82).

Furthermore, according to reviewer's suggestion, we performed flow cytometry analysis to assess the apoptosis of *E. coli*-infected primary hippocampal neurons pretreated with various apoptosis inhibitors including Bax channel blocker (inhibitor of Bax-mediated mitochondrial cytochrome c release), mitochondria inhibitor (iMac2, inhibitor of mitochondrial apoptosis-induced channel), pan-caspase inhibitor (Z-VAD-FMK) and caspase-3 inhibitor (Z-DEVD-FMK). Interestingly, we found that Bax channel blocker and iMac2, but not caspase inhibitors, were effective to attenuate the neuron apoptosis induced by *E. coli* (new Fig. S7C and D in the revised manuscript). These results suggested that the rapid neuron apoptosis in response to *E. coli* infection occurred through mitochondrial apoptotic pathway that was less dependent on caspase activity. Compatible with our findings, the caspase-independent neuronal apoptosis has been reported in primary cortical neurons treated with the β -hemolysin/cytolysin from Group B *Streptococcus* (*J Infect Dis.* 2011; 203: 393–400). These were incorporated to the revised manuscript in Page 14, Line 322-326. We hope these results could relieve reviewer's concern about the neuron apoptosis induced by *E. coli*.

Minor Points:

In Fig. 1D, the authors should provide more details of their nuclear marker 38F3.

Reply: The 38F3 indicated the antibody against fibrillarin, a component of ribonucleoproteins that was often used as a marker for nuclear fraction. We corrected the “38F3” to “fibrillarin” in Fig. 1D in the revised manuscript.

In Fig. 1G, the electron micrographs do not have the clarity to show that Caspr1 is localized to the luminal side of brain microvessels.

Reply: For better visualization, we adjusted the brightness and contrast of the electron micrograph in Fig. 1G. Furthermore, a zoom-in view of the positive immunoreactivity of Caspr1 (gold particles) at the luminal side of brain microvessels were placed on the right of Fig. 1G.

In Fig. 2, the authors use the number of mice with “positive CSF” to quantify the percent of mice in which the bacteria enters the BBB. Is it standard practice to not include the actual numbers of bacteria in the CSF (the way the CFUs are shown for bacteria levels in the blood)? Is there a threshold below or above which the CSF is considered negative or positive, respectively, for the bacteria?

Reply: Here we would like to explain the method for measurement of the percentage of mice with *E. coli* meningitis. The CSF of mice infected with rifampin-resistant *E. coli* strain were collected and cultured on agar plate containing rifampin (100 µg/ml) at 37 °C overnight, and then the number of bacterial colonies were counted to calculate CFU (colony-forming unit). The CSF specimen yielding bacterial colonies greater than 1 was defined as positive culture, whereas CSF specimen yielding no bacterial colony was defined as negative. In fact, so far, the lowest value we obtained from CSF culture is 4×10^3 CFU/ml, which means 4 bacterial colonies were observed from the plate of CSF culture. The mice with positive culture in CSF were recorded as successful penetration of bacteria through the BBB resulting in meningitis. As reviewer mentioned, it is a standard practice to calculate the percentage of mice with meningitis without mention the actual bacterial number in the CSF, which was utilized in our manuscript as well as in previous studies (Nat Commun. 2011;2:552, J Biol Chem. 2002;277:15607-12, J Clin Invest. 1992;90:897-905). To further relieve reviewer’s concern, we provided the original data of the bacterial numbers, calculated as CFU/ml, derived from CSF of mice in Fig. 2C as a table below.

Table. Analysis of *E. coli* meningitis occurrence in neonatal mice

Group	Mice #	Bacteria in blood (log CFU/ml of CSF)	Bacteria in CSF (CFU/ml)	Number of animals with positive CSF and meningeal inflammation	Meningitis occurrence rate	P value
Control	1	6.31	0	11	11/18 (61%)	
	2	6.60	1×10^5			
	3	4.48	2×10^4			
	4	4.70	4×10^4			
	5	6.28	5×10^4			
	6	6.26	0			
	7	6.54	1×10^6			
	8	4.30	0			
	9	4.88	0			
	10	4.72	1×10^4			
	11	4.73	4×10^3			
	12	5.48	5×10^5			
	13	5.20	0			
	14	4.70	2.5×10^5			
	15	6.35	6×10^3			
	16	6.11	0			

	17	5.31	0			
	18	6.23	7.5 X 10 ⁵			
						
Caspr1						
eKO	1	4.87	0	4	(4/16) 25%*	P<0.05
	2	6.51	0			
	3	5.49	0			
	4	4.88	0			
	5	6.51	1.5 X 10 ⁴			
	6	4.20	0			
	7	5.70	0			
	8	5.31	0			
	9	6.04	0			
	10	4.48	0			
	11	6.45	7.5 X 10 ⁴			
	12	6.30	0			
	13	5.58	5 X 10 ³			
	14	5.00	0			
	15	5.57	0			
	16	6.34	5 X 10 ⁵			

Reviewer #4 (Remarks to the Author):

This is an interesting paper and the data that Caspr1 acts a receptor for the entry of E. coli IbeA entry into HBMEC cells is pretty convincing as is the data to suggest that drugs that block this interaction have the potential to be developed into a new treatment strategy.

Reply: Thanks for reviewer's support and encouragement on our manuscript.

The data included in Figs. 4 and 6 is much less convincing and appears preliminary in nature. There are numerous comments below in regards to most of the figures, many reflect the limited information provided for many of the experimental conditions and can be addressed by modifying the text/figure legends. However, additional experiments would be needed to strengthen the interpretation of experiments presented in Figs. 4 and 6.

Reply: According to reviewer's comments, we provided the detailed information in the revised manuscript as reviewer suggested. Furthermore, we performed additional experiments to further support the results in Fig. 4 and Fig. 6. For details, please read the point-to-point responses provided below.

1. Fig.1

A. in the Y2H screen were any other potential interacting proteins identified? If so, what? How many colonies were screened? Were 4 distinct clones of CaspR isolated? Coordinates of clones? More details of results of screen should be included in methods or supplement.

Reply: According to reviewer's suggestion, the results of the yeast two-hybrid (Y2H) screen were provided as new Table S1 (attached below) in supplementary information in the revised manuscript. Regarding the clones of Caspr1 obtained from Y2H screen, we obtained 4 distinct clones (clone #40, #62, #65, #72) containing the coding sequence corresponding to the N-terminal LamG domain of Caspr1. As reviewer suggested, the coordinates of clones encoding Caspr1 were marked on the structure domains of Caspr1 in Fig. 3A and Fig. S3A.

Table S1 Results of yeast two-hybrid screen using IbeA as the bait protein

Clone #	Genbank accession #	Gene name
35, 66	NM_001101	Actin, beta
10	NM_004077	Citrate synthase
40, 62, 65, 72	NM_003632	Caspr1 (contactin associated protein 1)
2, 6, 20, 22, 24, 34, 50, 61	BC065494	DEAD/H (Asp-Glu-Ala-Asp/His) box polypeptide 12
19, 46, 47, 58	NM_000402	Glucose-6-phosphate dehydrogenase
33	NM_007260	Lysophospholipase II
3, 56	NM_015889	Mediator complex subunit 15
11	AY129569	Polymerase (DNA directed), delta 1, catalytic subunit
32	NM_001105570	Nudix hydrolase 19
4, 8, 15, 25, 26, 27, 38, 39, 67, 68, 69	NM_020850	RAN binding protein 10
14, 63	NM_133452	Ribonucleoprotein, PTB binding 1
29, 57	BC013878	Thimet oligopeptidase 1
41	NM_006086	Tubulin beta 3 class III
5	NM_004651	Ubiquitin specific peptidase 11

B. How are expts in 1B/C conducted? Is IbeA being expressed in the HEK and HBMEC cells? Or are cell lysates being run over GST- columns prebound with proteins of interest? This is unclear from both figure legend and text. If co-expressed, isn't it surprised that an interaction is detected between presumably cytosolic IbeA and Caspr1 _ as its interacting domain is presumably extracellular.

Reply: Here we would like to explain the experimental details of Fig. 1B and C. For Fig. 1B, the cell lysates from HEK293T cells transfected with full-length Caspr1 were incubated with Glutathione Sepharose 4B beads prebound with GST-IbeA or GST (as control), then the precipitated complex were subjected to Western blot to examine the binding of IbeA with exogenous expressed Caspr1 (Fig.1 B). For Fig. 1C, the cell lysates from normal HBMECs were incubated with Glutathione Sepharose 4B beads prebound with GST-IbeA, GST-OmpA or GST (the latter two served as controls), then the precipitated complex were subjected to western blots to assess the binding of IbeA with endogenous Caspr1 (Fig.1 C). These details were added to the results and the figure legend of Fig.1 B and C in the revised manuscript for easy understanding. Note that we did not co-express IbeA and Caspr1 in HEK293T cells or HBMECs in Fig. 1 and thereafter in our study.

C. In Fig. 1C, is anything known about the titer of bacteria in CSF? Is the titer in the CSF of the eKO lower than the control mice? If this is the case, would include this data.

Reply: The titer of bacteria in the CSF of mice with *E. coli* meningitis at 18 h post-infection have been recorded as qualitative data to determine whether meningitis occurred (Fig. 2C). In contrast, the titer of bacteria is inappropriate to assess the degree of meningitis due to the large variation of the values. From the attached graph below showing the bacteria titer in the CSF of mice with *E. coli* meningitis, the value is 248.2 ± 348.2 CFU/ml (mean \pm SD, n=11), exhibiting large variations, which was likely caused by the bacterial proliferation in the CSF during the 18 h period of time after bacterial inoculation. Thus, the titer of bacteria in the CSF of mice with *E. coli* meningitis was only used as qualitative data to determine the rate of meningitis occurrence.

Figure R1. The levels of bacteria in the CSF of wild-type mice with *E. coli* meningitis. *E. coli* (1×10^4 CFU) was injected subcutaneously into the wild-type mice. At 18 h post-infection, the blood samples were collected for bacteremia measurement and the CSF were collected and cultured to count the CFU as bacterial titer in CSF. The titer of bacteria in the CSF of mice with positive CSF culture were plotted.

D. The images shown in Fig. 1E suggest that most of Caspr1 is mostly perinuclear rather than membrane localized, yet the fractionations studies suggest that it is all located in the membrane. This is mentioned, but not really emphasized. I guess this might explain why the proteins co-IP when presumably co-expressed in HEK or HBMEC cells in Fig. 1B. (Also in these images is it surprising that most VE-cadherin appears to be nuclear?)

Reply: In Fig. 1D, we used Subcellular Protein Fractionation Kit (ThermoFisher Scientific) to isolate cytoplasmic, membrane, nuclear and cytoskeletal proteins from HBMECs and we found Caspr1 was exclusively localized in the membrane fraction. According to the manufacturer's instruction, the obtained membrane fraction includes not only plasma membrane, but also membranes from endoplasmic reticulum (ER), Golgi complex and mitochondria. In Fig. 1E, we observed that Caspr1 is localized at the plasma membrane of HBMECs, with intense

perinuclear staining. Thus, the perinuclear staining of Caspr1 in Fig. 1E likely indicated the presence of Caspr1 in ER because Caspr1, bearing mannose-rich N-glycans, was reported to be transported from ER to plasma membrane (J Biol Chem. 2003;278:48339-47, J Cell Biol. 2000; 149:491-502). These were added to the revised manuscript at Page 20, Line 460-468.

In addition, there might be some misunderstandings about Fig. 1B because we did not co-express IbeA and Caspr1 in the HEK293T cells. For Fig. 1B, the cell lysates from HEK293T cells transfected with full-length Caspr1 were incubated with Glutathione Sepharose 4B beads prebound with GST-IbeA or GST (as control), then the precipitated complex were subjected to Western blot to examine the binding of IbeA with exogenous expressed Caspr1 (Fig. 1 B).

Regarding the dominant staining of VE-cadherin in the nucleus in Fig. 1E, it might be caused by non-specific staining of the antibody against VE-cadherin. We repeated the experiments in Fig. 1E using new VE-cadherin antibody, and the results showed that the signal of VE-cadherin at the plasma membrane was improved whereas the signal in the nucleus was significantly reduced in the new Fig. 1E presented in the revised manuscript.

E. In Fig. 1D, details in method regarding adherence assay are very limited. Are the infected cells extensively washed before number of bacteria quantified? Presumably yes- but never mentioned.

Reply: When performing *E. coli* adherence assay, the infected HBMECs were washed thoroughly 3 times by RPMI 1640 media before quantification of bacteria. According to reviewer's comments, the detailed methods of bacterial adhesion assay (attached below) were added to the Experimental Procedure of the revised manuscript in Page 33, Line 774-780.

Bacterial adhesion assay

The *E. coli* were added to the confluent HBMECs cultured in 24-well plate with multiplicity of infection (MOI) of 100. The plates were incubated in the incubator (37 °C, 5 % CO₂, 95 % humidity) for 1.5 h to allow adhesion to occur. Then the cells were thoroughly washed 3 times with RPMI 1640 media to remove unbound bacteria, and were lysed with 0.5% Triton X-100. The lysates were plated on LB agar plates and cultured overnight to count the bacterial colony-forming unit (CFU) for quantifications.

Also, by quantifying all the bacteria present- how can they differentiate between adherent and invasive bacteria?

Reply: As reviewer pointed out, the quantification data derived from *E. coli* adherence assay includes both adhered and invaded bacteria. However, the containing of invaded bacteria cannot affect the conclusions draw from *E. coli* adherence assay because the invaded bacteria are ~ 1000 times less than the adhered bacteria. With normal HBMEC, the level of invaded bacteria measured from *E. coli* invasion assay is $1.50 \pm 0.56 \times 10^4$ CFU/ml, whereas the adhered bacteria measured from *E. coli* adherence assay is $1.6 \pm 0.6 \times 10^7$ CFU/ml. Additionally, it is challenging to remove the intracellular bacteria when quantifying bacteria that adhered to the host cells.

Thus, it is a standard practice to count both adhered and invaded bacteria in bacterial adherence assay which was utilized in our manuscript as well as in previous studies (Infect Immun. 2017;85. pii: e01069-16, PLoS Pathog. 2012;8(6):e1002733).

Lastly, the Y-axis label is confusing. From methods it appears that they are quantifying bacteria in the well, thus nothing is known regarding the % of HBMEC cells with adherent bacteria. To draw such a conclusion, they would need to image infected cells. Thus, at this point appears pre-mature to conclude that the bacteria are not impaired in adherence.

Reply: There might be some misunderstandings. Regarding the Y-axis of the data of *E. coli* adhesion assay, it is a relative value compared to the data of normal HBMECs in which the number of adhered bacteria was defined as 100 %. For a better presentation, we provided the absolute value of adhered *E. coli* with normal HBMECs ($3.2 \pm 1.2 \times 10^6$ CFU/well) in the figure legend of the revised manuscript. Note that the Y-axis here is not the percentage of cells with adhered *E. coli* as reviewer mentioned.

Regarding the reviewer's question about the percentage of cells with adhered bacteria, we performed immunostaining experiments to assess the percentage of HBMECs with adhered bacteria after co-incubation with *E. coli* for 1.5 h, the time of which is exactly the same with that in *E. coli* adhesion assays. The results (attached below) showed that 100 % of HBMECs were found to be attached with *E. coli* (n=61 cells, from 3 independent experiments). Similarly, in the HBMECs with Caspr1 knockdown, 100 % of the cells were adhered with *E. coli* (n=58 cells, from 3 independent experiments). These results indicated that Caspr1 is not involved in *E. coli* adhesion to brain endothelial cells, which is in line with the results of *E. coli* adhesion assays in Fig. 2F in the revised manuscript.

Figure R2. Analysis of the percentage of cells with adhered *E. coli*.

HBMECs were cultured on coverslips, and then incubated with wild-type *E. coli* with multiplicity of infection (MOI) of 100 for 1.5 h in the incubator (37 °C, 5 % CO₂, 95 % humidity). Then the cells were washed 3 times with RPMI 1640 media to remove unbound bacteria. The cells were fixed and subjected to immunostaining. The *E. coli* were stained

with *E. coli* antibody (green), and the cytoskeletal actin was showed with phalloidin to visualize the outline of the cells (red) (A). For quantifications, the percentage of cells with adhered *E. coli* were calculated (number of cells with adhered bacteria/total number of cells \times 100) (B). Data are from 3 independent experiments.

F. In Fig. 1E, as in Fig. 1D, Y-axis is not correctly labeled. Unless imaging cells cannot determine % OF HBMEC cells with intracellular bacteria. Rather the way assay is designed, the investigators are basically comparing the number of intracellular bacteria in the cells in presence and absence of *Caspr1*. This number reflects both the invasion and intracellular replication of bacteria, but tell us nothing about % of infected cells.

Reply: Regarding the Y-axis of the data of *E. coli* invasion assay, it is a relative value compared to the data of normal HBMECs in which the number of internalized bacteria was defined as 100 %. Note that the Y-axis here is not the percentage of cells that contained intracellular *E. coli* as reviewer mentioned. In the manuscript, we provided the absolute value of invaded *E. coli* with normal HBMEC in all the related figure legends.

The reviewer mentioned that the results from *E. coli* invasion assay may reflect both the invasion and intracellular replication of the bacteria. To relieve reviewer's concern, we performed experiments to monitor the intracellular replication (1, 2, 4 h post-infection) of *E. coli* after its internalization into HBMECs. Our results showed that the internalized *E. coli* did not replicate within the normal HBMECs as well as within the *Caspr1*-silenced HBMECs (attached below). These results excluded the possibility of intracellular replication of the invaded *E. coli* within HBMECs during the period (1.5 h) of *E. coli* invasion assays. In previous studies, the intracellular replication of Group B *Streptococci* (Infect Immun. 1997;65:5074-81) and meningitis isolate *E. coli* IHE3034 (Infect Immun. 1996;64:2391-9) were not observed in mammalian cells, although the intracellular replication of *Citrobacter freundii* (Infect Immun. 1999;67:4208-15) was reported, indicating that the intracellular replication of bacteria is limited to certain type of bacteria. Thus, the *E. coli* invasion assays used in our study are reliable approaches to assess the degree of *E. coli* invasion.

Figure R3. Intracellular replication of *E. coli* in HBMECs

HBMECs were seeded in 24-well plates until formation of confluent monolayers. The cells

were then incubated with wild-type *E. coli* with multiplicity of infection (MOI) of 100 for 1.5 h in the incubator (37 °C, 5 % CO₂, 95 % humidity). Then the cells were washed 3 times with RPMI 1640 media. After addition of gentamycin (100 µg/ml) to kill the extracellular bacteria, the 24-well plates were put back to the incubator for additional incubation for 1, 2 and 4 h. The cells were lysed with 0.5 % Triton X-100 and the lysates were plated on LB agar plates and cultured overnight to count the bacterial colony-forming unit (CFU) for quantifications. Values are from 3 independent experiments. Control, HBMEC stably transfected with empty vector. Caspr1 KO, stable HBMEC cell line with Caspr1 knocked out by CRISPR-Cas9 system.

2. Fig. 3

A. In Fig. 3A, would be helpful to include information regarding coordinates of the fragments of Caspr1 detected in Y2H plus provide information regarding the % similarity/homology shared by the 4 LamG domains.

Reply: In the new Fig. 3A and Fig. S3A in the revised manuscript, we marked the coordinates of the fragments of Caspr1 obtained from Y2H screen as reviewer suggested.

According to reviewer's suggestion, we further provided the sequence comparison results of the 4 LamG domains as new Fig. S3C in the revised manuscript. The results showed that the 4 LamG domains share 40 % similarity.

B. Fig. 3E, did the investigators confirm that Caspr1 delta 203-355 is stably expressed and correctly localized? If so, would state that this is the case.

Reply: The stable expression of Caspr1 Δ 203-355 construct in HBMECs was verified by western blot analysis in the bottom panel of Fig. 3E in our manuscript. Given that the Caspr1 Δ 203-355 construct contained a His-tag that fused with cDNA encoding Caspr1 Δ 203-355, the anti-His antibody was used in western blot to recognize the His-tag-fused Caspr1 Δ 203-355. The results showed that Caspr1 Δ 203-355 is stably expressed in HBMECs (bottom panel, Fig. 3E).

According to reviewer's suggestion, we performed immunostaining experiments to examine the cellular distribution of Caspr1 Δ 203-355, and we observed the localization of Caspr1 Δ 203-355 at the plasma membrane in the stably-transfected HBMECs with intense perinuclear staining (new Fig. S3B in the revised manuscript), which is similar to the localization pattern of endogenous Caspr1 in HBMECs.

C. Fig. 3, is IbeA delta 229-342 secreted by the E. coli and if so, at levels equivalent to WT IbeA? This needs to be shown to definitely demonstrate that the mutant strain defect is directly due to the inability of this region of IbeA to bind to and mediate uptake of the bacteria into the HBMEC cells.

Reply: We performed further experiments to assess the secretion of IbeA Δ 229-342 by *E. coli*

as reviewer suggested. The *ibeA*-deleted ZD1 *E. coli* strain was transfected with full-length *ibeA* and *ibeA* Δ 229-342, respectively. Then the *E. coli* were co-cultured with HBMECs and the supernatant were collected for western blot analysis. The bacterial pellet of the *E. coli* without co-incubation with HBMECs were loaded as positive controls. The results (new Fig. S3D in the revised manuscript) showed that IbeA Δ 229-342 could be secreted from *E. coli* after co-incubation with HBMECs and the secreted level of IbeA Δ 229-342 is similar to that of wild-type IbeA protein.

3. Fig. 4:

A. The reciprocal IPs in 4 A are not very convincing. Suggest weak interaction, if any. The immunoblots shown are not as convincing as the Coomassie gels shown in the supplement.

Reply: As reviewer mentioned, the interaction of Caspr1 with FAK was identified by the results of precipitation followed by mass spectrometry in Fig. S4A. Then in Fig. 4A, reciprocal immunoprecipitation were performed to further verify the interaction between Caspr1 and FAK. Note that the results (Fig. 4A) here are reflecting the endogenous association of Caspr1 with FAK, which may be the reason why the precipitated bands are not as evident as expected in overexpression system. Although the intensity of the precipitated protein is not very strong (top part of the images in Fig. 4A), it was sufficient to be visualized, indicating the positive interaction between Caspr1 and FAK. Moreover, the interaction of Caspr1 with FAK were consistently repeated in the following results in Fig. 4B and D (top part of the images).

B. In Fig. 4B, time is presumably in minutes- would clarify. What happens if look at later time points? Does data become more convincing?

Reply: The time unit is minutes in Fig. 4B. The text “5, 15, 30 (min)” was marked on the top of Fig. 4B to indicate the time points (5min, 15 min, 30 min).

Here we would like to explain the time points chosen in Fig. 4B. From the results of *E. coli* invasion assays, we knew that it takes at least 1 h for *E. coli* to invade into HBMEC. Therefore, we chose 5, 15 and 30-min time points, which is earlier than the *E. coli* internalization, to assess the activation of intracellular signaling pathways (including Caspr1-FAK interaction, FAK phosphorylation, etc.) that is required for *E. coli* internalization. Our results consistently showed that Caspr1-FAK interaction and FAK phosphorylation reached the peak at 15 min post-infection. As reviewer mentioned, there might be alterations of intracellular signaling later than 1 h post-infection, but these alterations are very likely associated with the secondary processes (such as intracellular survival and trafficking of *E. coli*) but not the initial *E. coli* invasion process, which is beyond the interest of the investigations in Fig. 4.

What are numbers shown in each lane- presumably quantification of interacting protein pulled down in WB? Or is it some type of ratio of the interacting protein to that of the pulled-down protein?

Reply: In Fig. 4B, those numbers indicated the ratio of the interacting proteins to that of precipitated proteins. For better presentation, those numbers were used to plot a new bar graph placed on the bottom of Fig. 4B in the revised manuscript. Furthermore, statistical analysis was performed and the P values with statistical significance were marked on the bar graph of Fig. 4B accordingly.

Is anything known regarding the percentage of infected cells at this time point? If it is low, then perhaps this explains weakness of detected interactions. Can the experiment be modified to promote adhesion of E. coli to the HBMEC cells? Presumably MOI of 100 is being used here? This should be clarified.

Reply: There might be some misunderstandings. From the results in Fig. 4B, we concluded that the interaction of Caspr1 with FAK in HBMECs was enhanced in response to *E. coli* infection and reached the peak at 15 min post-infection. Yet, we did not claim that the interaction of Caspr1 with FAK is correlated with the percentage of infected cells. It will be interesting to explore whether the interaction between Caspr1 and FAK is correlated with the percentage of infected HBMECs in future studies.

In addition, the reviewer questioned about the infection conditions used in Fig. 4B. Here we would like to explain that in Fig. 4B, we used multiplicity of infection (MOI) of 100 to achieve successful infection of HBMECs with *E. coli*. The primary reason why we use MOI of 100 is to ensure consistency, because MOI of 100 is used in all the bacterial invasion and adhesion assays with HBMECs in our manuscript. According to reviewer's suggestion, the information of MOI was added to the figure legend of Fig. 4B in the revised manuscript.

C. In Fig. 4F is it known whether the caspr1 delta C1 is correctly localized to the membrane? If not, the significance of data is unclear. The phosphorylation data in supplemental Fig. C is not very convincing.

Reply: According to reviewer's suggestion, immunostaining experiments were performed to examine the cellular distribution of Caspr1 Δ C, and we observed the localization of Caspr1 Δ C at the plasma membrane in HBMECs stably transfected with Caspr1 Δ C construct (new Fig. S4C in the revised manuscript), which is similar to the localization pattern of endogenous Caspr1 in HBMECs.

Furthermore, we quantified the band intensities of the western blot results in Fig. S4D in the revised manuscript and statistical analysis were performed. The results showed that the phosphorylation of cPLA2 and Src were increased upon *E. coli* infection, however, this increased phosphorylation was not affected by transfection with Caspr1 Δ C ($P > 0.05$, new Fig. S4D in the revised manuscript).

D. Fig. 4D. This expt is overinterpreted. Presumably bacteria invasion is markedly inhibited in the caspr1 delta C lines, thus Rac1 activation could be due to any downstream event associated

with bacteria invasion. Thus the conclusion on line 219 that *rac1* is associated with intracellular signaling of *Caspr1* is an overstatement.

Reply: According to reviewer's suggestion, we modified the conclusion as "Rac1 might be associated with intracellular signaling of Caspr1" in the revised manuscript in Page 11, Line 242.

E. Fig. 4I, can the E. coli invade the FAK1 depleted cells?

Reply: We performed *E. coli* invasion assay with HBMECs transfected FAK-specific siRNA, with HBMECs transfected with non-silencing siRNA as control. The results showed that FAK knockdown reduced the *E. coli* invasion into HBMECs (attached below). In line with our results, it has been reported that *E. coli* invasion into HBMECs were significantly reduced by transfection with FAK mutants, such as FAK(Phe397) mutant defective in autophosphorylation, FRAK mutant that lacking the amino-terminal end containing the kinase domain (Infect Immun. 2000;68:6423-30).

Figure R4. *E. coli* invasion assay with FAK-depleted HBMECs

HBMECs were transiently transfected with siRNA against FAK, with non-silencing siRNA as control. The knockdown effect were analyzed by Western blot (lower panel). Then the HBMECs were used for bacterial invasion assay (upper panel). Values are mean \pm SD from 3 independent experiment. **, $P < 0.01$. The absolute value of the intracellular bacterial CFU in control (HBMECs) was $3.3 \pm 1.6 \times 10^3$ CFU/well.

4. Fig. 5

A. Fig. 5C/D. Would include dose of peptide given SC in Fig. 5C. Presumably same amt as given in 5D.

Reply: The dosage of injected peptide in Fig. 5C and D were exactly the same (30 μ g/rats), and the dosage was added in the figure legend of the revised manuscript accordingly.

B. In Figure 5, is it thought that the peptide acts to both block as well as attenuate the degree to which mice develop meningitis? Are there differences in the appearance of the tissue section or CSF of mice treated with peptide that develop meningitis vs. those that do not? Are the sections in (E) from a mouse treated with peptide that developed meningitis (defined by positive CSF cultures). Similarly, how many of the three CSF samples shown in (F,G) were obtained from mice with evidence of meningitis – that received peptide.

Reply: Thanks for reviewer’s helpful comments and these prompted us to reconsider the results in Fig.5.

We would like to explain the original experimental design for Fig. 5E, F and G. Our results in Fig. 5D indicated that the injection of Caspr1(203-355) peptides reduced the passage of *E. coli* through the BBB in neonatal rats indicated as the reduced rate of positive culture of CSF. Then we performed histological analysis of the brain sections to assess the meningeal inflammation. We found that the meningeal inflammation was consistently observed in the rats with positive CSF culture and was absent in the rats with negative CSF culture, which is in line with the results in Fig. 2C. Therefore, in Fig. 5E, the representative histological images were presented to show the normal cerebral meninges of the rats injected with Caspr1(203-355) peptide with negative CSF culture indicating negative meningeal inflammation. For comparison, the representative histological images of the control rats with positive CSF culture were presented in Fig. 5E to indicate the occurrence of meningeal inflammation. In brief, the histological images in Fig. 5E are representative images to provide the further evidence to support the reduced occurrence of meningitis in response to Caspr1(203-355) injection. For same reason, in Fig. 5F and G, the polymorphonuclear leukocytes (PMN) numbers and glucose concentration in the CSF from 3 representative rats with negative CSF culture injected with Caspr1(203-355) peptides were presented, to compare with that from 3 representative rats with positive CSF culture in control group.

During the revision process, we realized that the results in Fig. 5E, F and G may cause misunderstandings. For a better presentation, we modified the header “No. of animals with positive CSF” to “No. of animals with positive CSF and meningeal inflammation” in the table of Fig. 5D in the revised manuscript. In addition, the results of Fig. 5F and G, which were somewhat redundant, were removed to avoid misunderstandings.

The reviewer’s additional question is that whether there are differences in the degree of meningitis in the rats injected with Caspr1(203-355) peptides that still developed to meningitis compared to the control rats with meningitis. To answer this question, all the rats with meningitis in each group were included to analyze the meningeal inflammation, PMN numbers and glucose concentration in the CSF. The results showed that the meningeal thickness, the levels of PMN numbers and glucose concentration in CSF were at similar levels in the rats injected with Caspr1(203-355) peptides that developed to meningitis (n=6), compared to the rats with meningitis in control group injected with GST alone (n=14) (Fig. S5B, C and D). These results suggested that when *E. coli* meningitis occurred, the subcutaneous injection with

Caspr1(203-355) peptides may have little effect on the degree of meningitis. Thus, we made an amendment to the conclusion draw from Fig. 5 as follows: “These results suggested that when *E. coli* meningitis occurred, the subcutaneous injection with Caspr1(203-355) peptides may have little effect on the degree of meningitis. Thus, subcutaneous injection of Caspr1(203-355) peptides specifically act on the initial entry step of bacteria into the CNS, i.e, the penetration of bacteria through the BBB, to reduce the occurrence of *E. coli* meningitis.”. These modifications were incorporated into the revised manuscript in Page 19-20, Line 450-459.

How many mice were studied in H (include in figure legend). Does the statement on line 254 that these mice had “experimental E. coli meningitis” mean that upon death, all the mice in Fig. H had positive CSF cultures? Or does it mean that all mice infected with doses of bacteria that survival was prolonged when mice were infected with normally lethal doses of bacteria?

Reply: There are 14 rats for each group in the Fig. 5H (changed to Fig. 5F in the revised manuscript) which was indicated in the figure legend.

Here we would like to explain the details of the survival experiments in Fig. 5F in the revised manuscript. All the neonatal rats were infected with same dose of *E. coli* (4×10^2 bacteria/rat) and the survival times of the rats were recorded daily until 7 days. The dosage used in survival curve experiments is based on the medium lethal dose (LD50) of *E. coli* in neonatal rats, which was calculated as 360 ± 80 CFU. In addition, for the dead rats, it is not possible to collect the blood and CSF specimen for the assessment of meningitis due to tissue liquefaction. According to reviewer’s comments, we corrected the statement as “Furthermore, we found injection of Caspr1(203-355) peptide resulted in prolonged survival of neonatal rats infected with certain dose of *E. coli* ($P < 0.05$, Fig. 5F)” in the revised manuscript in Page 12, Line 273-275.

5. Figure 6

(A) Were only three told images quantified? Were they all from the same mouse? Seems that multiple images of multiple mice should be imaged before drawing any conclusions

Reply: There might be some misunderstandings about Fig. 6A. The images of Fig. 6A were from 9 mice in total (n=9) for each group from 3 independent experiments. These details were added to the figure legend of Fig. 6A in the revised manuscript.

(B) How many neurons were imaged? How often were bacteria visualized? Is invasion dependent on IbeA? How many bacteria/neuron visualized? Need to provide some type of quantification to support the significance of the presented image. At what times post-invasion was the cell shown processed for imaging? Same conditions as in C?

Reply: The Fig. 6B is a representative image from 48 infected neurons containing *E. coli* as revealed by electron microscopy (EM) analysis. In the revised manuscript, we provided 5 additional representative EM images for neurons infected with wild-type *E. coli* as new Fig. S6A.

According to reviewer's suggestion, we quantified the results in Fig. 6B. The percentage of bacteria-containing neurons is 5.57 ± 0.86 %, and the number of bacteria within the neurons were 2.00 ± 1.48 in primary neurons infected with wild-type *E. coli*. Furthermore, we performed EM experiments to assess the percentage of bacteria-containing neurons infected with *ibeA*-deficient ZD1 strain, and we found that the percentage was significantly reduced to 0.19 ± 0.11 % ($P < 0.0001$, new Fig. 6C, with 5 representative EM images in new Fig. S6A).

Regarding the time of harvest for EM experiments in Fig. 6B and C, the neurons were infected with *E. coli* for 60 min. The reason is that we found that the neurons at 60 min post-infection is more suitable for EM, whereas the neurons at 90 min post-infection are too fragile for EM examination. To further relieve reviewer's concern, we performed *E. coli* invasion assays of neurons to compare the internalized *E. coli* derived from different incubation times. The results showed that the levels of invaded *E. coli* in neurons at 60 min post-infection was slightly higher than that at 90 min, but showed no statistical significance (attached below). The results indicated that the conclusions draw from the EM results with neurons infected with *E. coli* (Fig. 6B and C) are in line with the results from *E. coli* invasion assays with neurons (Fig. 6D) in our manuscript.

Figure R5. *E. coli* invasion assays with primary rat hippocampal neurons with different incubation times

Primary rat hippocampal neurons were seeded in 24-well plates for 4 days. The neurons were then incubated with wild-type *E. coli* with multiplicity of infection (MOI) of 100 for indicated times (30, 60, 90 min) in the incubator (37 °C, 5 % CO₂, 95 % humidity). Then the cells were washed 3 times with Neurobasal media followed by addition of gentamycin (100 µg/ml) to kill the extracellular bacteria. The neurons were then lysed with 0.5 % Triton X-100 and the lysates were plated on LB agar plates and cultured overnight to count the bacterial colony-forming unit (CFU) for quantifications. Values are from 3 independent experiments.

(C) How many hippocampal neurons were infected in each experiment? Presumably thousands? If so seems that invasion is very inefficient. For example if 1,000 cells are infected at an MOI of 100 – that would mean that 100,000 bacteria were initially in a well – yet only 20 bacteria are isolated post-gent treatment. And, note- gent only added after 90 minutes- so the bacterial in the wells like doubled at least two times. Thus, if the bacteria do invade the hippocampal

cells, they do so with very low efficiency. However, if the infected cells are dying, as suggested by the supplemental data, than the number of quantified intracellular bacteria will be difficult to determine, because if the bacteria are released by the dead cells they will be killed by the gent. In addition, gent will be able to penetrate into the dead cells and hence kill intracellular bacteria. The investigators might test whether numbers of intracellular bacteria increase in the presence of an apoptosis inhibitor.

Reply: The reviewer's major concern here is that the efficiency of *E. coli* to invade neurons is low. To relieve reviewer's concern, we compared the invasion efficiency of *E. coli* into several different types of cells, including HBMECs, primary rat hippocampal neurons, cervical cancer HeLa cells, neutrophil-like HL-60 cells (differentiated myeloid leukemia cells induced by dimethylsulfoxide), murine macrophages RAW264.7 cells. The results (Fig. S6E, attached below), with the Y-axis defined as invaded *E. coli* (CFU) per well (2×10^5 host cells), showed that different cells exhibited distinct *E. coli* invasion efficiency. As expected, the large amount of *E. coli* within RAW264.7 indicated the high rate of phagocytosis in macrophages to uptake *E. coli*. We noticed that the invasion of *E. coli* into cervical cancer HeLa cells is barely detectable, which is in line with the property of the cervical epithelial cells that are less prone to be infected by meningitic *E. coli* than brain endothelial cells. Regarding the neurons, the invasion efficiency of *E. coli* into neurons is similar to that in HBMECs, both apparently higher than that in HeLa cells, which is in line with the neurotropic property of meningitic *E. coli* strain. These data indicated that *E. coli* can invade into neurons with moderate efficiency. These were incorporated into the revised manuscript in Page 18, Line 428-431.

Figure S6E. *E. coli* invasion assays with different mammalian cells HBMECs, primary rat hippocampal neurons, cervical cancer HeLa cells, neutrophil-like HL-60 cells (differentiated myeloid leukemia cells induced by dimethylsulfoxide) and murine macrophages RAW264.7 cells were seeded in 24-well plates for 4 days. The cells were then

incubated with wild-type *E. coli* with multiplicity of infection (MOI) of 100 for 90 min in the incubator (37 °C, 5 % CO₂, 95 % humidity). Then the cells were washed 3 times with culture media followed by addition of gentamycin (100 µg/ml) to kill the extracellular bacteria. Then the cells were lysed with 0.5 % Triton X-100 and the lysates were plated on LB agar plates and cultured overnight to count the bacterial colony-forming unit (CFU) for quantifications. Values are from 3 independent experiments.

The reviewer's additional concern is that the number of invaded *E. coli* may be difficult to determine. To our knowledge, there are multiple factors that can cause alterations of the intracellular invaded *E. coli*. For example, longer incubation will increase the internalized bacteria due to increased invasion of *E. coli*. On the contrary, the cell apoptosis induced by prolonged incubation will reduce the intracellular bacteria due to killing of bacteria by gentamycin as reviewer mentioned. Despite these, to assess the degree of bacterial invasion into neurons, one have to choose a specific time point post-infection to conduct the *E. coli* invasion assay for the comparison of the bacterial invasion rate between different groups. Given that the infection time is 90 min in *E. coli* invasion assay with HBMECs, for consistency, here the neurons at 90 min post-infection were used for the *E. coli* invasion assay of neurons in Fig. 6D, F, G and H. Our results showed that, with MOI of 100, $11.30 \pm 1.92 \times 10^3$ CFU/ml *E. coli* per 2×10^5 neurons were collected from the cytoplasm of primary hippocampal neurons (Fig. 6D). From these results, we concluded that *E. coli* are able to invade into the cytoplasm of neurons.

(D) Is anything known about Caspr1 expression in the hippocampal neurons? Are those the labeled cells in Fig. 2B that were not discussed?

Reply: The immunostaining and western blot analysis with Caspr1 antibody revealed that Caspr1 is expressed in the primary cultured mouse hippocampal neurons (left panel of Fig. 6E, bottom panel of Fig. 6F in the revised manuscript). Further western blot results identified the expression of Caspr1 in the primary rat hippocampal neurons (bottom panel of Fig. S6D in the revised manuscript). These findings demonstrated the presence of Caspr1 in hippocampal neurons. In line with our results, the expression of Caspr1 in hippocampal neurons has been reported in previous study (J Biol Chem. 2012; 287: 6868–6877). Regarding the labeled cells in the brain parenchyma of wild-type mice in Fig. 2B, these positive stained cells indicated the expression of Caspr1 in the brain cortex because the presented images were from brain cortex.

REVIEWERS' COMMENTS:

Reviewer #1 (Remarks to the Author):

The authors have adequately answered my previous concerns.

Reviewer #2 (Remarks to the Author):

Comments reviewer 2 to the rebuttal and the revised manuscript:

General:

There are still quite a few occasions where articles are missing in the text. This still needs to be corrected.

Furthermore, the wording needs to be more cautious, not stating 'hope' as a 'fact' (see line 31).

Figure 1C and D:

When you compare Figure 1C with 1D you notice that GST-IbeA migrates differently compared to the markers. This needs to be explained. The arrow signs are different.

Line 94 ff:

Figure 1E: The new anti-VE-cadherin antibody still gives a heavy stain in the nucleus which raises doubts about its quality or stability. Hence, the authors should add an explanation to this end in the legend of Figure 1E and point to Figure 1D where in the cell fractionation no VE-cadherin shows up in the nucleus.

Line 100 and Fig. 1G:

Well, ok – but the EM images are still rather unsatisfactory.....

Line 111:

ok

Line 116:

ok

Line 137:

ok

Line 141:

ok

Line 150:

ok

Line 154:

ok

Line 156 etc.:
ok

Line 157/160:
ok

Line 167:
In the rebuttal the authors state Caspr1 Δ 203-255 however in Figure S3B they state Caspr1 Δ 203-355.
Which is correct ?

Line 168:
ok

Line 173/174:
ok

Line 194 ff and Figure 4:
ok

Line 235 ff:
ok

Line 261:
Well, ok

Line 851:
ok

Fig. 4E and Fig 5:
ok

Reviewer #3 (Remarks to the Author):

The authors have made a good effort to characterize the ability of E.coli to induce apoptosis in neurons. In particular, the different methods the authors use to show that Caspr1 is necessary for this apoptosis in neurons are good additions to the manuscript. The characterization of the specific pathway of apoptosis induced is still weak but there is already a lot of data in this manuscript and the pathway of apoptosis induced is not a central focus of this paper. The authors should however define the various reagents used to characterize this pathway (in Fig. S7D) in their methods section. For example, which Bax inhibitor is used in this study (at what concentration), etc.

Reviewer #4 (Remarks to the Author):

The authors have done an excellent job of responding to my concerns and have put together an impressive story.

Responses to Reviewer's comments

Reviewer #1 (Remarks to the Author):

The authors have adequately answered my previous concerns.

Reply: Thanks for reviewer's support on our manuscript.

Reviewer #2 (Remarks to the Author):

General:

There are still quite a few occasions where articles are missing in the text. This still needs to be corrected.

Reply: Thanks for reviewer's careful reading and we corrected all the missing articles according to reviewer's suggestion.

Furthermore, the wording needs to be more cautious, not stating 'hope' as a 'fact' (see line 31).

Reply: We modified the sentence at Line 31 according to reviewer's suggestion as follows: "Our results indicate that *E. coli* exploits Caspr1 as a host receptor for penetration of BBB resulting in meningitis, and that Caspr1 might be a useful target for prevention or therapy of *E. coli* meningitis".

Figure 1C and D:

When you compare Figure 1C with 1D you notice that GST-IbeA migrates differently compared to the markers. This needs to be explained. The arrow signs are different.

Reply: There might be some misunderstandings about the position of migrated GST-IbeA in Fig. 1D. Actually, the GST-IbeA used in Fig. 1D is exactly the same as the GST-IbeA used in Fig. 1C. To avoid any misunderstandings, we labeled the molecular weight of the protein marker in Fig. 1D. In addition, the arrow signs of the band of GST-IbeA in Fig. 1D were modified to be consistent with the signs in Fig. 1C as reviewer suggested.

Line 94 ff:

Figure 1E: The new anti-VE-cadherin antibody still gives a heavy stain in the nucleus which raises doubts about its quality or stability. Hence, the authors should add an explanation to this end in the legend of Figure 1E and point to Figure 1D where in the cell fractionation no VE-cadherin shows up in the nucleus.

Reply: According to reviewer's suggestion, we added an explanation in the figure legend of Fig. 1E as follows: "The expression of VE-cadherin in the nuclear fraction was undetectable in (D), thus we consider the fluorescence signals of VE-cadherin in the nucleus might be caused by non-specific staining".

Line 100 and Fig. 1G:

Well, ok – but the EM images are still rather unsatisfactory.....

Reply: From the zoom-in windows on the right panel of Fig. 1G, the positive labeling of the luminal expression of Caspr1 were clearly presented. We believe these could relieve the reviewer's concern about Fig. 1G.

Line 111:

ok

Line 116:

ok

Line 137:

ok

Line 141:

ok

Line 150:

ok

Line 154:

ok

Line 156 etc.:

ok

Line 157/160:

ok

Reply: Thanks for reviewer's careful reading.

Line 167:

In the rebuttal the authors state Caspr1 Δ 203-255 however in Figure S3B they state Caspr1 Δ 203-355. Which is correct ?

Reply: It should be Caspr1 Δ 203-355. Thanks for reviewer's kind reminder.

Line 168:

ok

Line 173/174:

ok

Line 194 ff and Figure 4:

ok

Line 235 ff:

ok

Line 261:

Well, ok

Line 851:

ok

Fig. 4E and Fig 5:

ok

Reply: Thanks for reviewer's comments and careful reading of the manuscript.

Reviewer #3 (Remarks to the Author):

The authors have made a good effort to characterize the ability of E.coli to induce apoptosis in neurons. In particular, the different methods the authors use to show that Caspr1 is necessary for this apoptosis in neurons are good additions to the manuscript. The characterization of the specific pathway of apoptosis induced is still weak but there is already a lot of data in this manuscript and the pathway of apoptosis induced is not a central focus of this paper.

Reply: Thanks for reviewer's support on our manuscript.

The authors should however define the various reagents used to characterize this pathway (in Fig. S7D) in their methods section. For example, which Bax inhibitor is used in this study (at what concentration), etc.

Reply: According to reviewer's suggestion, the detailed information of the inhibitors used in Fig. S7D were provided in the figure legend of Fig. S7D as follows" When indicated, the neurons were pretreated with apoptosis inhibitors including Bax channel blocker (10 μ M, Tocris), mitochondrial inhibitor iMAC2 (10 μ M, Tocris), pan Caspase inhibitor Z-VAD-FMK (10 μ M, Tocris) and Caspase3 inhibitor Z-DEVD-FMK (10 μ M, Tocris), respectively, for 30 min before bacterial infection".

Reviewer #4 (Remarks to the Author):

The authors have done an excellent job of responding to my concerns and have put together an impressive story.

Reply: Thanks for reviewer's great support on our manuscript.